# 🎧 JavisGPT: A Unified Multi-modal LLM for Sounding-Video Comprehension and Generation

**Kai Liu**[1,2]*, **Jungang Li**[3]*, **Yuchong Sun**[4]*, **Shengqiong Wu**[2], **Jianzhang Gao**[4],
**Daoan Zhang**[5], **Wei Zhang**[6], **Sheng Jin**[7], **Sicheng Yu**[8], **Geng Zhan**[9], **Jiayi Ji**[2],
**Fan Zhou**[1], **Liang Zheng**[10], **Shuicheng Yan**[2], **Hao Fei**[2]†, **Tat-Seng Chua**[2]

[1]ZJU, [2]NUS, [3]HKUST(GZ), [4]RUC, [5]UR, [6]HZCU, [7]NTU, [8]SMU, [9]USYD, [10]ANU

**Project:** https://JavisVerse.github.io/JavisGPT-page

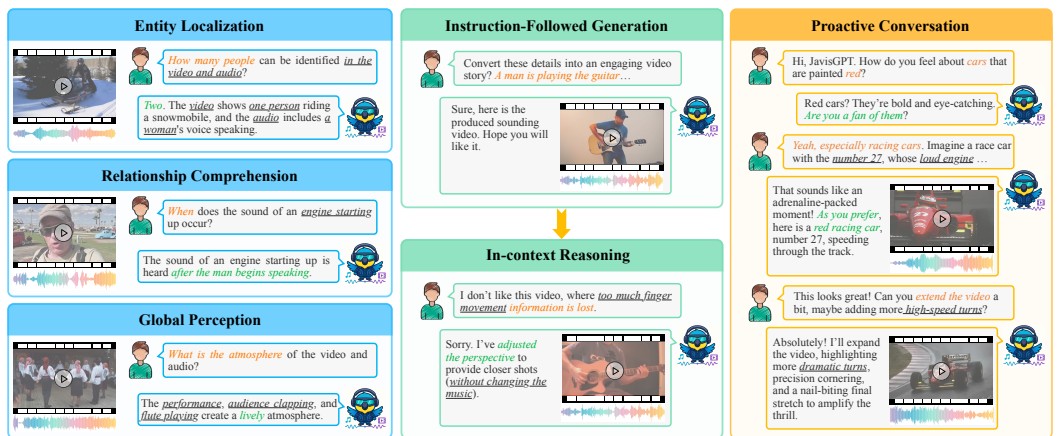

Figure 1: JavisGPT supports multi-level synchronized audio-video content (*i.e.*, sounding video) comprehension (**left**), and simultaneously complex instruction-based audio-video generation, interleaved in-context generation (**middle**), and multi-turn proactive conversations (**right**).

## Abstract

This paper presents **JavisGPT**, the first unified multimodal large language model (MLLM) for joint audio-video (JAV) comprehension and generation. JavisGPT has a concise encoder-LLM-decoder architecture, which has a SyncFusion module for spatio-temporal audio-video fusion and synchrony-aware learnable queries to bridge a pretrained JAV-DiT generator. This design enables temporally coherent video-audio understanding and generation from multimodal instructions. We design an effective three-stage training pipeline consisting of multimodal pretraining, audio-video fine-tuning, and large-scale instruction-tuning, to progressively build multimodal comprehension and generation from existing vision-language models. For instruction tuning, we construct `JavisInst-Omni`, a high-quality instruction dataset with over 200K GPT-4o-curated audio-video-text dialogues that cover diverse and multi-level comprehension and generation scenarios. On JAV comprehension and generation benchmarks, our experiments show that JavisGPT outperforms existing MLLMs, particularly in complex and temporally synchronized settings.

---

*Equal contribution. Work done during Kai Liu's visiting period at NUS. Email: kail@zju.edu.cn
†Corresponding author. Email: haofei7419@gmail.com

39th Conference on Neural Information Processing Systems (NeurIPS 2025).

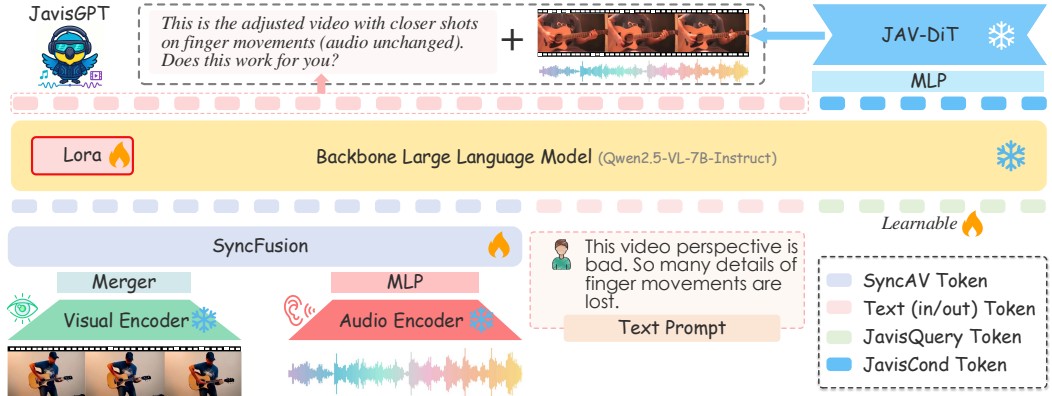

Figure 2: **Overall architecture of JavisGPT**, which can perceive and produce videos and sounds simultaneously. The input video and audio are encoded and fed into the **SyncFusion** module, and the resulting synchronized audio-video representation, together with the text tokens and learnable `JavisQueries` are then passed to the LLM backbone. During decoding, the yielded `JavisCond` embeddings are used to align the LLM intents to the semantic conditional space of downstream JAV-DiT, enabling high-quality and synchronized sounding-video generation.

## 1 Introduction

Joint understanding and generation of multiple modalities has recently sparked increasing interest [71, 80, 21, 72, 86, 50]. Existing efforts generally focus on image-text frameworks [96, 84, 73, 51, 11], or treat video and audio understanding and generation as separate modalities [90, 81, 42].

We introduce **JavisGPT**, a bespoke MLLM for high-quality, unified comprehension and generation in joint audio-video (JAV) tasks, which has clear application potential in avatar chatbots, sounding movie curation, and music-driven video analysis. JavisGPT is a holistic system that accepts as input separate audio and videos, synchronized sounding videos, and user text prompts. After comprehending and reasoning over these input signals, it produces either textual responses and/or synchronized sounding videos. Our architecture supports a wide range of JAV scenarios, including question-answering in comprehending sounding audios, generation of sounding audios following user instructions, upon in-context reasoning, and interleaved text-audio-video generation in multi-turn user dialogues (ref. Fig. 1). As a comprehensive and state-of-the-art system, we highlight the design of JavisGPT below (with detailed discussions with related works provided in Sec. 2).

**Model architecture.** JavisGPT uses an encoder-LLM-decoder architecture (ref. Fig. 2), with Qwen2.5 [87] as the LLM backbone. The visual encoder is inherited from Qwen2.5-VL [5], and the audio encoder is based on BEATs [10]. Audio and video features, along with user prompts and learnable JavisQuery tokens, are passed to the LLM. To enable fine-grained spatiotemporal alignment, we propose a dedicated SyncFusion module that fuses audio and video representations into synchronized SyncAV tokens for unified comprehension. At the output stage, the LLM generates textual responses along with JavisCond tokens, which encode contextual semantics and serve as conditioning inputs for a pretrained JAV-DiT generator [39]. We choose JavisDiT for its generation quality and flexibility, and incorporate hierarchical JavisQueries to provide spatiotemporal priors, further enhancing synchronization in audio-video generation.

**Training strategies.** To effectively adapt the vision-language backbone to JAV tasks, we design a three-stage training pipeline: (1) *MM-PreTrain*: introducing the audio input branch for comprehension and preliminarily aligning the output embeddings of LLM with the condition space of JavisDiT; (2) *AV-FineTune*: enhancing synchronized audio-video comprehension and generation via SyncFusion and hierarchical JavisQueries, respectively; and (3) *MM-InstTune*: enables generalizable instruction following and multimodal reasoning via large-scale instruction tuning across text, video, and audio.

**Dataset collection.** To facilitate instruction tuning, we construct `JavisInst-Omni`, a large-scale, diverse, and high-quality instruction dataset covering both comprehension and generation. The dataset contains 200K dialogue trajectories involving tightly interleaved text, audio, and video, simulating complex single- and multi-turn interactions. All samples are annotated via GPT-4o and manually

verified for quality. The instructions span a wide range of task formats, including QA, captioning, instruction-following generation, and in-context multimodal reasoning.

We compare JavisGPT with existing JAV-like LLMs on various JAV benchmarks, where JavisGPT yields impressive performance across all scenarios. Moreover, for audio-video joint generation, JavisGPT also significantly outperforms existing methods, producing high-quality and diversified sounding videos. We reiterate our main points below:

- We present JavisGPT, the first unified MLLM for sounding-video comprehension and generation.
- We present its architecture design, including the audio-video synchronization mechanism, its multi-stage tuning strategy, and a large-scale instruction dataset for JAV LLMs training.
- Evaluation indicates the effectiveness of JavisGPT in JAV comprehension and generation.

## 2   Related Work

**Multimodal comprehension.** Recently, MLLMs have significantly advanced many multimodal fields, such as the modeling of image, video, and audio contents [5, 12, 18, 60, 22, 20, 79, 55]. Early approaches, such as Flamingo [4] and BLIP [33, 34], focused on combining text with *static* visual inputs. Recent research extended these efforts to *dynamic* vision and audio modalities, such as PandaGPT [67], Video-LLaMA [91, 14], UnifiedMLLM [36], and MiniOmni [85], which incorporate temporal and auditory perception and reasoning. While these MLLMs achieve promising performance, they typically treat modalities (*e.g.*, audio and video) as independent inputs from separate channels, relying on concatenation or simple feature padding for integration. This strategy overlooks fine-grained spatio-temporal interactions between modalities, compromising their effectiveness in complex scenarios where multimodal synchronization is required, such as JAV.

**Multimodal generation.** Compared to research efforts for multimodal content comprehension, relatively limited emphasis has been placed on generative capabilities for MLLMs, particularly for scenarios involving multiple interdependent modalities, such as JAV [31, 55]. For audio-video generation, existing MLLMs like HuggingGPT [64], NExT-GPT [81], and UnifiedIO-2 [42] adopt a pipeline-based framework, where individual audio and video generation components are sequentially executed. Although such methods achieve reasonable results, they suffer from issues like error propagation and desynchronized outputs due to the lack of joint modeling.

**Unified comprehension and generation.** Early approaches such as AV2AV [15] and MultiDialog [52] tackled talking-head generation and speech-driven dialogue, but were limited by simple architectures or the absence of textual input/output. While recent MLLMs aim to unify multimodal comprehension and generation, most still focus on single modalities, lacking effective modeling of cross-modal interactions, particularly the audio-video synchronization. For instance, Video-Salmonn [68, 70] and Meerkat [16] use separate encoders for audio and video with little attention to alignment. Similarly, NExT-GPT [81] and AnyGPT [90] support multimodal generation but fail to synchronize audio and video due to separate decoders. To address these limitations, we propose JavisGPT, which integrates a spatio-temporal alignment mechanism and unified feature learning, advancing the state of the art in joint audio-video (JAV) modeling.

## 3   Architecture

JavisGPT leverages Qwen2.5-VL-7B-Instruct [5] as backbone, incorporating with SyncFusion and JavisQueries to perform joint comprehension and generation. Sec. 3.1 demonstrates how Javis-GPT extends to audio perception and utilizes SyncFusion to explicitly capture spatio-temporal synchrony in input sounding videos. Sec. 3.2 introduces how JavisGPT integrates with a pretrained JAV generator [39] and enhance the generation synchrony through hierarchical JavisQueries.

### 3.1   Synchrony-aware Audio-Video Comprehension

**Modality-specific encoding for video and audio.** Following recent works [14, 68, 70], we first adopt the vision encoder from Qwen2.5-VL [5] to encode the video inputs to yield dense video representation $\mathbf{V} \in \mathbb{R}^{T_v \times (H \times W) \times C}$, where $T_v$ denotes the number of video frames, $(H, W)$ are the spatial dimensions, and $C$ is the feature dimensionality. Meanwhile, we introduce a parallel audio encoding branch built upon BEATs [10] to extract dense audio representations $\mathbf{A} \in \mathbb{R}^{T_a \times M \times C}$,

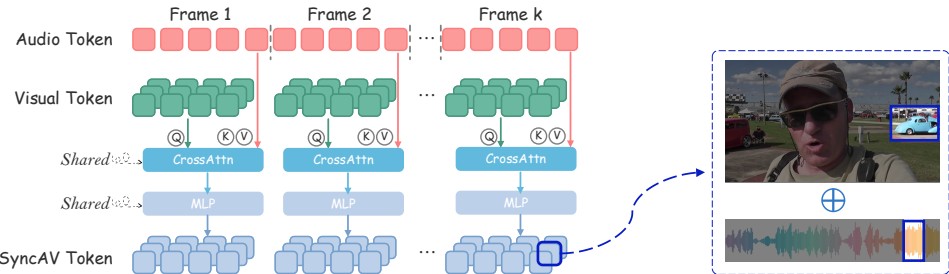

Figure 3: **Mechanism of SyncFusion.** (1) **Left**: We align temporally-segmented audio tokens with corresponding video frames by using cross-attention to merge audio clues into visual patches, so as to capture spatiotemporal synchrony explicitly. (2) **Right**: Each resulting SyncAV token $e_{i,j}^t \in \mathbb{R}^C$ represents a sounding event occurring within the $i$-th row, $j$-th column visual patch of $t$-th frame.

where $T_a$ is the temporal length of the audio features, $M$ is the number of frequency bins, and $C$ is the feature dimension. Specifically, the raw audio waveform is first converted into a 2D mel-spectrogram, comprising $T_a$ time frames and $M = 128$ frequency bins. Each audio frame corresponds to a 25 ms window with a 10 ms overlap, enabling high-resolution temporal encoding. Finally, a two-layer MLP $\phi_\theta^a$ is applied to project audio features into the LLM token embedding space, serving as a bridge for JavisGPT to understand audio content.

**Synchronized audio-video fusion.** Unlike previous MLLMs that straightforwardly concatenate video and audio features together [14, 68] or simply interleave audio-video frames [70, 86], JavisGPT introduces a SyncFusion module $\psi^{av}$ to explicitly model the spatio-temporal synchrony in sounding videos. As illustrated in Fig. 3, we first temporally align the audio and video streams by evenly dividing the audio representations to match the number of video frames, transforming the audio features from $\mathbf{A} \in \mathbb{R}^{T_a \times M \times C}$ to $\mathbf{A}' \in \mathbb{R}^{T_v \times (r \times M) \times C}$, where $r = \lceil T_a/T_v \rceil$, with zero-paddings applied if needed. Then, a shared cross-attention module is employed to inject audio cues into the frame-wise video representations, followed by a two-layer MLP to generate *SyncAV* representations. The length of *SyncAV* representations matches that of the vanilla extracted video representations, ensuring compatibility with the MRoPE positional encoding used in the Qwen2.5-VL [5]. Each resulting token $e_{i,j}^{t_v} \in \mathbb{R}^C$ encodes a localized audio-visual event occurring at the $i$-th row, $j$-th column visual patch of frame $t_v$. By integrating audio semantics into visual features, the SyncFusion module enhances representational expressiveness and achieves precise spatio-temporal synchronization. This is crucial for downstream tasks requiring fine-grained audio-video understanding.

## 3.2 Spatio-Temporally Synchronized Audio-Video Generation

**Equipping backbone MLLM with DiT generator.** Drawing inspiration from Pan et al. [51], we introduce $N$ learnable query embeddings $Q^c$ to bridge the hidden state space of the LLM with the semantic condition space of the DiT generator. As demonstrated in Fig. 4, when JavisGPT encounters the special token `<|av_start|>` during the NTP process, the $N$ learnable tokens are immediately inserted into the input sequence. These tokens are processed through the causal attention layers of the LLM, after which a two-layer MLP $\phi_\theta^c$ are applied to project the $N$ hidden states $\mathbf{h} = \mathbb{R}^{N \times C}$ into the condition embedding space of the DiT, yielding $\hat{\mathbf{c}} = \mathbb{R}^{N \times D}$, which serve as semantic conditions for generating synchronized sounding videos. Unlike MetaQueries [51], which completely replaces the original conditions $\mathbf{c}$ with $\hat{\mathbf{c}}$ and require full fine-tuning of the DiT, the proposed JavisGPT retains the original DiT parameters by freezing the generator and instead aligns $\hat{\mathbf{c}}$ with the original T5-XXL [59] text encoder with alignment loss $\mathcal{L}_{align}^c = \| \hat{\mathbf{c}} - \mathbf{c} \|_2$ [81]. This design greatly simplifies training and reduces computational cost while maintaining effective conditioning for generation, and Sec. D provides further discussions with other generative MLLMs.

**Synchrony-enhanced audio-video generation.** To further strengthen the spatio-temporal synchrony of generated sounding videos, we follow Liu et al. [39] to additionally design $N'$ learnable query tokens $Q^s$ to estimate spatio-temporal prior (ST-prior) embeddings $\hat{\mathbf{s}}$ for auxiliary conditions. As shown in Fig. 4, the semantic condition $\hat{\mathbf{c}}$ a coarse description of the overall audio-visual event (*e.g.*, 'a car starts its engine and leaves the screen'), while the ST-prior condition $\hat{\mathbf{s}}$ captures fine-grained synchrony. Specifically, the spatial prior governs where events occur and disappear (*e.g.*, 'the car is in the top-left corner of screen'), whereas the temporal prior determines when they begin and end

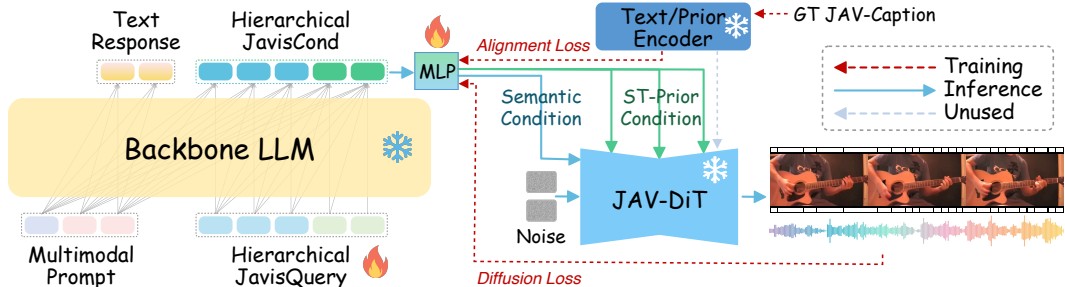

Figure 4: **Instruction-followed and synchronized audio-video generation.** We use learnable queries to gather useful information from all the audio, video, and text inputs, and use a two-layer MLP to map the hidden states from LLM to the conditional space of DiT. The hierarchical semantic and spatiotemporal prior conditions further enhance the synchrony in generated sounding videos. Alignment loss and diffusion loss are integrated to reduce optimization difficulty and training cost.

(*e.g.*, 'sound starts at 2s, exits at 7s, fades at 9s'). To compute $\hat{s}$, we employ a separate MLP projector $\phi_\theta^s$ to map the final-layer representations of the prior query tokens into the ST-prior condition space, aligning them with the ST-prior encoder used in JavisDiT [39]. Finally, we combine the alignment loss between the predicted ST-prior embeddings $\hat{s}$ and the ground-truth ST-priors $s$ with the denoising objective, jointly optimizing both to strengthen the audio-video synchrony:

$$\mathcal{L}_{comp} = \parallel \hat{\mathbf{c}} - \mathbf{c} \parallel_2 + \parallel \hat{\mathbf{s}} - \mathbf{s} \parallel_2 + \parallel \epsilon - \hat{\epsilon}(\mathbf{a}_t, \mathbf{v}_t, \hat{\mathbf{c}}, \hat{\mathbf{s}}, t) \parallel_2 \triangleq \mathcal{L}_{align}^c + \mathcal{L}_{align}^s + \mathcal{L}_{diff}^{c,s} , \quad (1)$$

where $\hat{\epsilon}(*)$ is the DiT model that predicts the noise/velocity added to video $\mathbf{v}$ and audio $\mathbf{a}$ at timestamp $t$ with (estimated) text condition $\hat{\mathbf{c}}$ and st-prior condition $\hat{\mathbf{s}}$, and $\epsilon$ is the ground-truth. We follow Liu et al. [39] to use rectified flow [40] as the scheduler.

## 4 Training and Datasets

This section presents the three-stage training pipeline designed to efficiently adapt the visual-language backbone MLLM [5] into our proposed JavisGPT for joint comprehension and generation of sounding videos. A brief summary of the training process is demonstrated in Tab. 1, with detailed training configurations provided in Sec. B.2.

Table 1: **Summary of the training pipeline**. 'MM-PT' denotes *multimodal pre-training*, 'AV-FT' refers to *audio-video fine-tuning*, and 'MM-InstTune' means *multimodal instruction-tuning*.

| Stage | Tasks | Trainable Modules | # Parameters | Training Objective |
|---|---|---|---|---|
| MM-PT | A→T, T→AV | $\phi^a, Q^{c,s}, \phi^{c,s}$ | 239.95M | $\mathcal{L}_{ntp} + \mathcal{L}_{align}$ |
| AV-FT | AV→T, T→AV | $\psi^{av}, Q^{c,s}, \phi^{c,s}, \Psi_{lora}^{LLM}$ | 654.68M | $\mathcal{L}_{ntp} + \mathcal{L}_{align} + \mathcal{L}_{diff}$ |
| MM-InstTune | A/V/AV+T→T+AV | $\phi^a, \psi^{av}, Q^{c,s}, \phi^{c,s}, \Psi_{lora}^{LLM}$ | 717.09M | $\mathcal{L}_{ntp} + \mathcal{L}_{align} + \mathcal{L}_{diff}$ |

### 4.1 Pre-training for Basic Video-Audio Comprehension and Generation

The first stage equips JavisGPT with basic AV comprehension and generation skills by extending Qwen2.5-VL to audio and aligning it with the JavisDiT generator.

**Audio input alignment learning.** As described in Sec. 3.1, we optimize the only parameters of the two-layer MLPs $\phi^a$, which are applied to project the audio features (encoded by BEATs [10]) to the input token embedding space of LLM. Specifically, following the setup of VideoLLaMA2 [14], we train $\phi^a$ on approximately 600K audio-text pairs using audio captioning and audio-based question-answering tasks by minimizing the next-token prediction loss $\mathcal{L}_{ntp}$ on the generated text responses.

**Preliminary audio-video generation alignment learning.** In this part, we learn $N$ learnable query embeddings $Q^c$ along with the projection network $\phi^c$ to map the LLM's hidden states into the condition space of the DiT generator, which is originally modeled by a T5-XXL encoder [59]. Specifically, The learnable query embeddings $Q^c$ aggregated contextual information from the LLM

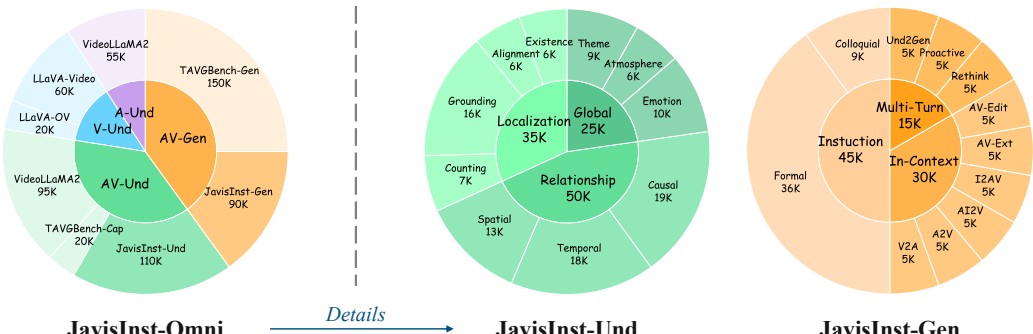

Figure 5: (1) **Left**: We curate a large-scale, diverse, and balanced cross-modal instruction-tuning dataset (`JavisInst-Omni`) from multiple sources. (2) **Right**: Within `JavisInst-Omni`, `JavisInst-Und` and `JavisInst-Gen` are specifically developed for multi-level audio-video comprehension and generation with synchrony-awareness. Details are presented in Sec. C.

is Specifically, we use 1.5M sounding video captions from TAVGBench [46] as input prompts, each concatenated with the $N$ queries, which serve to aggregate contextual information from the LLM. Then, the resulting hidden states $Q^c$ corresponding to these queries are projected by $\phi^c$ to obtain the estimated condition embeddings $\hat{\mathbf{c}}$. To accelerate and stabilize the semantic alignment between the LLM and DiT, we minimize an alignment loss $\mathcal{L}_{\text{align}}$ between the predicted conditions $\hat{\mathbf{c}}$ and the reference conditions $\mathbf{c}$ encoded by the frozen T5-XXL from the same captions. This pre-alignment phase is crucial for providing semantically meaningful initialization, which in turn facilitates subsequent training of fine-grained audio-video synchronization.

## 4.2 Fine-tuning for Synchronized Audio-Video Comprehension and Generation

Further, the model is fine-tuned to enhance its ability to capture fine-grained audio-video synchrony. LoRA [27] ($r = 128, \alpha = 256$) is integrated with the LLM backbone to strengthen the adaptation.

**Synchrony-aware audio-video comprehension.** This part focuses on optimizing the SyncFusion module $\psi^{av}$ to strengthen spatio-temporal alignment of the audio-video representations. The module is trained on the sounding-video caption task, where sounding videos are provided as input, and the LLM is required to generate visual and acoustic descriptions. We use around 360K audio-video-text triplets from TAVGBench [46], with supervision provided by the NTP loss.

**Synchrony-aware audio-video generation.** In this stage, we further fine-tune the semantic and st-prior queries $Q^c, Q^s$ along with the output projectors $\phi^c, \phi^s$, aiming to improve the synchrony of generated sounding videos. Training is conducted by minimizing the loss defined in Equation (1), allowing the model to better align semantic intent with fine-grained spatio-temporal dynamics during generation. We reuse the collected 360K audio-video-text triplets from TAVGBench for this stage.

## 4.3 Instruction-tuning for Interleaved Comprehension and Generation

As an interactive chatbot, JavisGPT is expected to not only be equipped with instruction-following capabilities but also improve its ability to reason over user intent for more generalized and user-friendly interaction. To this end, we first construct a large-scale, diverse instruction tuning dataset, named `JavisInst-Omni`, to enable flexible and interleaved audio-video comprehension and generation. The overall dataset statistics are depicted in Fig. 5.

**JavisInst-Omni dataset.** First, to maintain model capacity of comprehending single video and audio modalities, we sample diverse QA instances from existing resources: 55K audio QA samples from VideoLLaMA2 [14], 60K video QA samples from LLaVA-Video-178K [95], and 20K image QA instances from LLaVA-OneVision [32]. Then, to enhance joint audio-video comprehension, except for 95K audio-video QA instances from VideoLLaMA2 [14] and 20K audio-video caption samples from TAVGBench [46], we additionally construct `JavisInst-Und` for multi-level synchrony-aware comprehension. Finally, to strengthen synchronized sounding video generation, except for 150K text-to-audio-video instances from TAVGBench [46] to maintain JAVG capability, we further curate `JavisInst-Gen` to handle various conversation scenarios. Sec. C provides more details.

Table 2: **Comparison on audio, video, and audio-video comprehension**. We use the ActivityNet [6], Perception [53], and MVbench [35] test sets for video comprehension, the ClothoAQA [38] and TUT2017 [49] test sets for audio comprehension, and the AVQA [89], MU-AVQA [41], and AVSD [2] test sets for synchronized audio-video understanding. JavisGPT exhibits competitive performance across all benchmarks. We also report the data (#Samples) used to adapt a backbone VLM for joint audio-video comprehension. **Best** and secondary results are marked **bold** and underline respectively.

| Model (7-8B) | #Samples | Video | | | Audio | | Audio-Video | | |
|---|---|---|---|---|---|---|---|---|---|
| | | ActivityNet | Perception | MVbench | ClothoAQA | TUT2017 | AVQA | MU-AVQA | AVSD |
| *Video-LLM* | | | | | | | | | |
| Video-LLaVA [37] | - | 56.5 | 67.9 | 58.6 | - | - | - | - | - |
| LLaVA-NeXT [93] | - | 53.5 | 48.8 | 46.5 | - | - | - | - | - |
| LLaVA-OV [32] | - | 55.6 | 57.1 | 56.7 | - | - | - | - | - |
| Qwen2-VL [76] | - | 57.4 | 62.3 | 67.0 | - | - | - | - | - |
| Qwen2.5-VL [5] | - | - | **70.5** | 69.6 | - | - | - | - | - |
| InternVL2.5 [12] | - | **58.9** | 68.9 | **72.0** | - | - | - | - | - |
| *Audio-LLM* | | | | | | | | | |
| Qwen-Audio [17] | - | - | - | - | 57.9 | 64.9 | - | - | - |
| Qwen2-Audio [18] | - | - | - | - | 60.9 | 73.7 | - | - | - |
| *Audio-Video-LLM* | | | | | | | | | |
| NExT-GPT [81] | 1.9M | 21.5 | 33.7 | 27.9 | 30.9 | 6.4 | 25.3 | 19.3 | 30.8 |
| UnifiedIO-2 [42] | 9.2B | 23.2 | 34.7 | 30.4 | 31.4 | 9.1 | 61.2 | 34.1 | 16.5 |
| Macaw-LLM [44] | - | - | - | - | - | - | 78.7 | 31.8 | 34.3 |
| AV-LLM [65] | - | 47.2 | - | - | - | - | 78.7 | 45.2 | 52.6 |
| VideoLLaMA [91] | - | 12.4 | - | - | - | - | 80.9 | 36.6 | 36.7 |
| VideoLLaMA2 [14] | 1.9M | 50.2 | 51.4 | 54.6 | 65.1 | 78.4 | - | 79.2 | 57.2 |
| VideoLLaMA2.1 [14] | 1.9M | 53.0 | 54.9 | 57.3 | 66.3 | 77.3 | - | 80.9 | 57.2 |
| Qwen2.5-Omni [86] | - | 57.2 | 70.4 | 66.9 | **68.0** | 78.3 | 91.5 | 79.9 | **62.8** |
| **JavisGPT (Ours)** | 1.5M | 58.1 | 70.2 | 68.4 | 67.3 | **82.1** | **93.8** | **82.1** | 62.2 |

**Joint instruction-tuning across modalities.** To enable JavisGPT supports diverse tasks shown in Fig. 1, we jointly fine-tune all newly introduced modules, including input/output projectors $\phi^a, \psi^{av}, \phi^c, \phi^s$, learnable queries $Q^c, Q^s$, and the LoRA adapter $\Psi_{lora}^{LLM}$, on diversified cross-modal instruction-tuning instances from our proposed `JavisInst-Omni` dataset.

## 5 Contribution Statement

This work is the result of a large and comprehensive project that aims to provide the community with an open-source tool and platform for joint understanding and generation of sounding video, which is the first of its kind. While most of the individual network components are known in the literature, it takes significant trials and errors to find a feasible plan, and connect and align them to form a state-of-the-art and yet concise system. In particular, the proposed three-stage training pipeline has proven useful for gradually improving model performance. In addition, the proposed `JavisInst-Omni` dataset represents a very rich mixture of single- and multi-modality data, with the majority being single-turn and the rest multi-turn. We believe the architecture, training methods, and datasets offer a very competitive baseline and have significant value for future data-centric and model-centric studies.

## 6 Experiments

This section presents the experimental results to evaluate the effectiveness of our proposed JavisGPT. For a fair comparison, we train JavisGPT following the three-stage pipeline described in Sec. 4, with each stage running for one epoch. We use zero-shot evaluation for downstream tasks including audio-video comprehension and generation using their official evaluation protocols. Additional implementation and training details can be found in Sec. B.

### 6.1 Comparison with the State of the Art

**Multimodal comprehension.** As shown in Tab. 2, our method achieves performance on par with advanced models on *video/audio understanding*, demonstrating strong unimodal capabilities. It maintains visual comprehension strength of Qwen2.5-VL [5] backbone (70.2 *vs.* 70.5 on Perception [53])

Table 3: **Comparison on audio-video generation.** Results are reported on JavisBench-mini [39]. We achieve stronger generation performance than standalone DiT models and unified MLLMs.

| Method | AV-Quality | | | Text-Consistency | | | | AV-Consistency | | | AV-Synchrony |
|---|---|---|---|---|---|---|---|---|---|---|---|
| | FVD↓ | KVD↓ | FAD↓ | TV-IB↑ | TA-IB↑ | CLIP↑ | CLAP↑ | AV-IB↑ | CAVP↑ | AVHScore↑ | JavisScore↑ |
| *DiT* | | | | | | | | | | | |
| MM-Diff [60] | 2311.9 | 12.2 | 27.5 | 0.080 | 0.014 | 0.181 | 0.079 | 0.119 | 0.783 | 0.109 | 0.070 |
| JavisDiT [39] | 248.9 | 1.9 | 7.6 | 0.273 | 0.145 | 0.307 | **0.382** | **0.203** | **0.799** | 0.181 | 0.153 |
| *MLLM* | | | | | | | | | | | |
| NExT-GPT [81] | 1463.2 | 6.1 | 7.0 | 0.215 | 0.132 | 0.182 | 0.371 | 0.067 | 0.732 | 0.061 | 0.038 |
| UnifiedIO-2 [42] | 1597.4 | 6.4 | **6.7** | 0.231 | 0.118 | 0.292 | 0.357 | 0.106 | 0.747 | 0.094 | 0.053 |
| **JavisGPT (Ours)** | **243.6** | **1.8** | 7.6 | **0.274** | **0.146** | **0.309** | 0.380 | 0.202 | 0.797 | **0.185** | **0.157** |

and even slightly outperforms Qwen2.5-Omni [86] on MVBench [35] (video) and TUT2017 [49] (audio). For *audio-video synchronized understanding*, JavisGPT surpasses both Qwen2.5-Omni and VideoLLaMA2.1 [14] on AVQA [89] and MU-AVQA [41], achieving top results on both benchmarks. We attribute this to the spatio-temporal alignment capability of the proposed SyncFusion and our synchrony-aware training strategy. Notably, those results are obtained with fewer training samples (1.5M in total), highlighting its strong data efficiency.

**Multi-modal generation.** Tab. 3 compares the text-to-audio-video generation performance of JavisGPT with existing methods. JavisGPT significantly outperforms NExT-GPT [81] and UnifiedIO-2 [42] in AV quality, semantic consistency, and AV synchrony. These gains largely stem from the explicit synchrony modeling through the spatio-temporal prior projector, coupled with a carefully designed training strategy. In addition, JavisGPT slightly outperforms the base generator of JavisDiT [39], which we attribute to the stronger semantic understanding and encoding capacity of the LLM backbone [5], leading to more precise condition representations, ultimately improving the quality of generated sounding videos.

**Interleaved comprehension and generation.** As no existing benchmark directly measures joint understanding and generation of sounding videos, we fill this gap by carefully curating a new evaluation set comprising 100 multi-turn QA dialogues covering four types of interleaved understanding-generation tasks. We employ five unbiased volunteers to conduct human evaluation over several dimensions, covering both textual and audio-video outputs. Full setup and detailed analysis are provided in Sec. E.1 The average results are reported in Fig. 6, where our method consistently outperforms the two competing methods with joint comprehension-generation capabilities by a large margin. Notably, we observe (1) NExT-GPT [81] frequently refuses to respond or generates noisy audio-video outputs, severely impacting *instruction-following* and *generation quality*; (2) UnifiedIO-2 [42] is inferior to *context reasoning* and *proactive thinking*, failing to connect user history (*e.g.*, preferences) with current instructions.

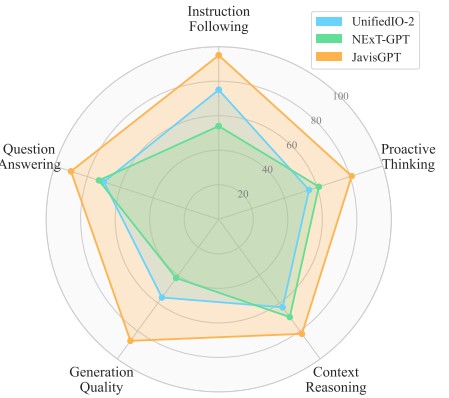

Figure 6: **Human evaluation on interleaved conversation.** JavisGPT significantly outperforms UnifiedIO-2 and NExT-GPT.

## 6.2 Further Analysis

**Effectiveness of SyncFusion on audio-video comprehension.** To assess the impact of synchronization strategies, we train models solely on the AV understanding subset of JavisInst-Omni and evaluate their performance across three AV comprehension benchmarks. In Tab. 4, we report the accuracy, the number of encoded AV tokens and per-sample inference latency. Results show that the interleaving strategy used in Video-Salmonn2 [69] and Qwen2.5-Omni [86] fails to outperform simple concatenation [37] and introduces additional latency (over 2× slowdown) due to memory-discontinuous tensor operations. The Q-former [34, 62, 81], despite offering faster inference, suffers from optimization instability [12, 5] and yields substantially lower performance. In contrast, our

Table 4: **Comparing SyncFusion with other options for AV-synchronization**. The SyncFusion module can effectively and efficiently capture the audio-video synchrony.

| AVSync | AVQA | MU-AVQA | AVSD | #Tokens↓ | Latency↓ |
|---|---|---|---|---|---|
| Concat | 93.3 | 80.7 | 61.3 | 3.5K | 246ms |
| Interleave | 93.3 | 80.6 | 61.6 | 3.5K | 555ms |
| BiCrossAttn | 93.2 | 80.6 | 61.1 | 3.5K | 263ms |
| Q-Former | 71.4 | 54.7 | 56.3 | *0.2K* | 182ms |
| **SyncFusion** | **93.4** | **81.4** | **62.0** | **2.0K** | **224ms** |

Table 5: **Ablation on training stages**. We compare training without audio-video fine-tuning (w/o AV-FT), without multimodal pretraining (w/o MM-PT), and with all stages (w/ All).

| Stage | Quality | | Consistency | | | Synchrony |
|---|---|---|---|---|---|---|
| | FVD↓ | FAD↓ | TV-IB↑ | TA-IB↑ | AV-IB↑ | JavisScore↑ |
| w/o AV-FT | 473.2 | 9.6 | 0.252 | 0.089 | 0.152 | 0.069 |
| w/o MM-PT | 306.7 | 8.4 | 0.267 | 0.132 | 0.178 | 0.135 |
| w/ All | **243.6** | **7.6** | **0.274** | **0.146** | **0.202** | **0.157** |

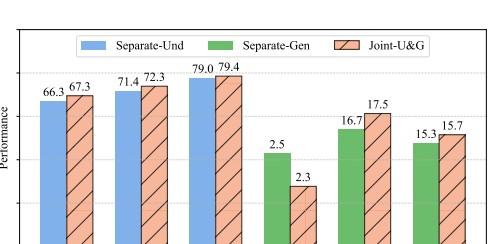

Figure 7: **Comparison on joint and separate training of understanding and generation**. Joint training generally leads to better performance, especially on the generation side.

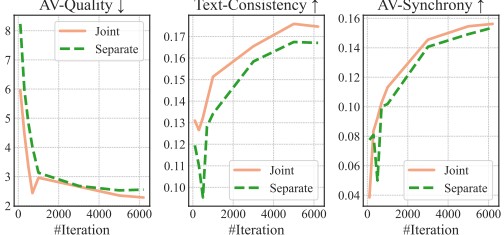

Figure 8: **Generation performance vs. training iterations**. We observe that joint training consistently improves generation quality, instruction following (text consistency) and synchronization.

proposed SyncFusion achieves strong AV understanding performance while reducing both token count and inference latency, showing a more balanced trade-off between performance and efficiency.

**Impact of the three training stages on high-quality AV generation.** Tab. 5 analyzes the contribution of each training stages to AV generation performance. We first evaluate the model after MM-Pretrain (Sec. 4.1, without AV-FineTune) and find that merely aligning with DiT's original text encoder [59] alone is insufficient to effectively bridge the LLM and DiT, resulting in poor generation quality (*e.g.*, a low JavisScore [39] of 0.069). Next, we skip pretraining and only apply diffusion loss for AV-Finetune (Sec. 4.2, without MM-Pretrain). While this significantly improves generation quality, consistency, and synchrony, training exhibits instability in early stages. To ensure convergence, we had to reduce the learning rate from 1e-4 to 1e-5, which ultimately compromises final performance. The best results are obtained by combining MM-Pretrain and AV-Finetune in our progressive training paradigm, confirming the necessity of staged adaptation for effective audio-video generation.

**Joint training improves both understanding and generation.** Fig. 7 compares the performance of comprehension and generation tasks under separate *vs.* joint training, with each score averaged over the corresponding test sets (*e.g.*, 'V-Und' is the average accuracy over ActivityNet [6], Perception [53], and MVBench [35]). Results show that joint training consistently outperforms separate training in both understanding and generation tasks, indicating a mutual enhancement effect. The improvement is more substantial on the *generation side*.

To better understand this effect, Fig. 8 presents the evolution of generation metrics across three dimensions. As the number of generation training samples increases, the inclusion of comprehension data leads to steady gains in both AV-Quality and Text-Consistency. It suggests that semantic understanding of AV content enhances the quality of condition embeddings used for generation. This trend also aligns with findings in MetaQueries [51] for image tasks. However, AV synchrony shows only marginal improvement (+0.04), possibly indicating the need for a more unified mechanism bridging comprehension and generation. We further discuss this in Sec. A.

**Case study: instruction-followed and synchrony-aware comprehension and generation.** Fig. 9 presents a qualitative comparison with UnifiedIO-2 [42] and NExT-GPT [81] in a two-round dialogue setting. In the first round, the models are asked to identify the source of a gunfire sound. UnifiedIO-2 gives a general response ('a semi-automatic weapon'), while NExT-GPT hallucinates an explosion of a flower. In contrast, our model accurately grounds the sound in a video game scene, showing better spatio-temporal grounding and scene understanding. In the second round, the models are asked

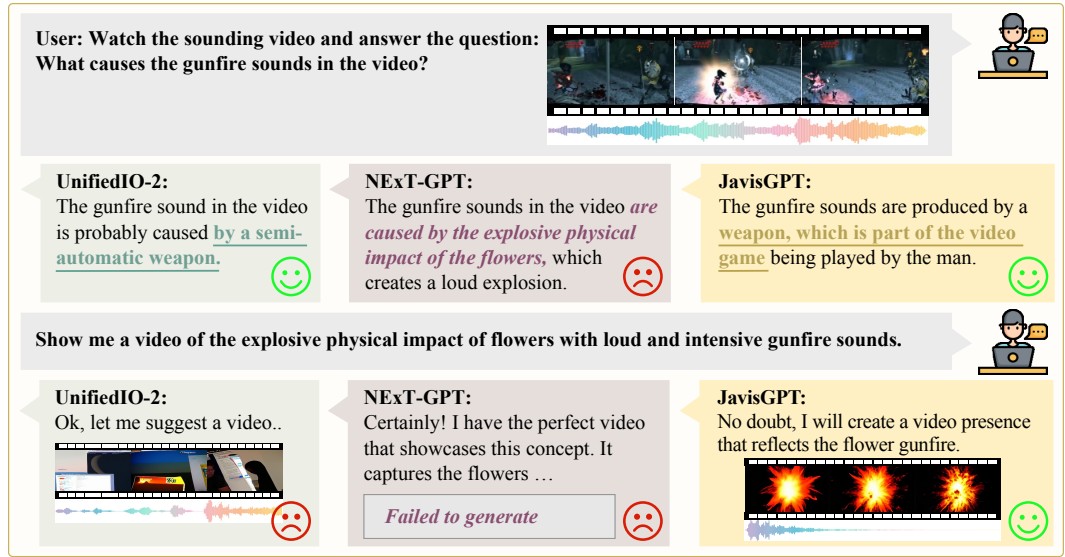

Figure 9: **Case study on joint audio-video understanding and generation.** (1) *Round-I* evaluates the instruction-based AV understanding; (2) *Round-II* validates on context-based AV generation. JavisGPT can faithfully understand and generate high-quality synchronous audio-video content.

to generate a gunfire-related video. UnifiedIO-2 produces irrelevant key frames, NExT-GPT fails to generate, while our JavisGPT successfully synthesizes coherent and synchronized audio-visual content. More examples are presented in Sec. F.

## 7  Conclusion and Discussion

We propose **JavisGPT**, a unified MLLM for synchronized audio–video comprehension and generation. JavisGPT adopts an encoder–LLM–decoder architecture with a SyncFusion module for spatiotemporal alignment and integrates a pretrained JavisDiT generator via hierarchical condition embedding. A three-stage training pipeline and the large-scale `JavisInst-Omni` dataset enable complex audio–video–text interactions. Extensive experiments demonstrate state-of-the-art performance on both unimodal and synchronized AV tasks, with interesting observations on mutual enhancement on both understanding and generation capabilities. We believe JavisGPT offers a strong foundation for future research in synchronized media generation.

**Discussion 1: extended capabilities for sounding video comprehension and generation**. Our Javis-GPT can be further enhanced by: (1) Enabling speech input/output by replacing BEATs with advanced models like Whisper or WavTokenizer [10, 57, 28, 86]; (2) Achieving fine-grained, controllable audio–video generation/editing via additional conditioning in JAV-DiT [29]; (3) Supporting complex instruction-based generation through improved LLM reasoning and contextual conditioning [78].

**Discussion 2: ultimate framework for joint audio-video comprehension and generation**. While encoder–LLM–decoder architectures with continuous inputs and diffusion decoders are well established [90, 81, 42], unified end-to-end autoregressive models using both discrete and continuous tokens show strong synergy in understanding and generation [77, 82, 73, 75]. Extending these from text–image to text–audio–video modeling holds great promise.

## Acknowledgement

This research/project is supported inpart by the National Research Foundation, Singapore, under its National Large Language Models Funding Initiative (AISG Award No: AISG-NMLP-2024-002), in part by Fundamental Research Funds for the Central Universities, and in part by Zhejiang Provincial Natural Science Foundation of China under Grant No. LDT23F01013F01. Any opinions, findings, and conclusions or recommendations expressed in this material are those of the author(s) and do not reflect the views of the National Research Foundation, Singapore.

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

In the **appendix**, we present discussion in limitation and societal impact (Sec. A,) details in model implementation and evaluation (Sec. B), construction details of JavisInst-Omni (Sec. C), discussion on architecture designs (Sec. D), supplementary investigations on JavisGPT (Sec. E), and more quantitative case studies on JavisGPT (Sec. F).

# A   Potential Limitation and Societal Impact

## A.1   Limitation and Future Work

While our JavisGPT represents a pioneering effort in synchronized audio-video understanding and generation, the field itself is still in its early stages. In this section, we discuss the current limitations of our method within the context of its time and outline several directions for future work.

**Architecture.** JavisGPT adopts a straightforward encoder–LLM–decoder architecture. While effective, as briefly mentioned in Sec. 7, this design presents two core inconsistencies: (1) *Misaligned training objectives*. The comprehension tasks rely on next-token prediction (NTP) loss [5] for textual outputs, whereas the generation tasks use diffusion loss [39] (with alignment loss) for audio-visual outputs. These distinct objectives lead to inconsistent optimization signals for the shared LLM backbone. (2) *Asymmetric input–output modeling*. For multimodal content (in this case, audio and video), the input relies on VAE-based continuous embeddings [5], while the output is conditioned through a query-based interface [51]. The connection between the two is indirect: the generation queries can "see" the embeddings from the comprehension stage, allowing understanding to enhance generation; however, the reverse is not true, as the generation queries are not part of the comprehension input. To achieve a truly unified MLLM for audio-video understanding and generation, it may be beneficial to draw inspiration from recent advances in unified modeling for image-based multimodal tasks [77, 82, 75] and adapt them to the audio-video domain. Sec. D further elaborates on possible architectural directions for unifying comprehension and generation, as well as improving spatiotemporal synchronization, which we hope can provide valuable insights for future research in the community.

**Scalability**. For the sake of rapid development and verification, our current implementation is built upon a 7B-scale backbone LLM [5] and trained on a limited collection of publicly available datasets. As such, the scalability of our approach has not been fully explored. With access to larger computational resources, scaling the backbone LLM to 70B-size or beyond, along with training on trillions of multimodal tokens, may lead to significant performance improvements and unlock further potential in joint audio-video modeling.

**Alignment.** In the current post-training stage, JavisGPT is only instruction-tuned to acquire basic instruction-following and simple reasoning capabilities. To further enhance the model's generalization ability, reinforcement learning (RL) presents a promising direction for post-training. Specifically: (1) On the comprehension side, RL can transform the base JavisGPT into a thinking model [25, 74, 88] capable of handling complex reasoning tasks in real-world scenarios. (2) On the generation side, RL can substantially improve the quality, textual consistency, and audio-video synchrony of the generated content [58, 63]. We leave this direction as an important avenue for future exploration.

## A.2   Societal Impact

JavisGPT enables synchronized audio-video comprehension and generation with language, supporting applications in education, accessibility, and interactive media. However, the same capabilities also raise important societal concerns. The model could be misused to synthesize misleading or harmful audio-visual content, contributing to the spread of misinformation or impersonation (e.g., deepfakes). If trained on sensitive or unfiltered data, it may also reproduce biased or private content, leading to fairness and privacy risks. Moreover, as generation models become more accessible, the barrier to misuse may decrease.

To mitigate these risks, responsible deployment practices are essential, including controlled access (e.g., through APIs), content watermarking, dataset auditing, and development of detection tools. While JavisGPT is released as a research prototype, we highlight the need for continuous evaluation of its broader impact as the field progresses.

# B JavisGPT Model Details

## B.1 Model Architecture

We propose JavisGPT, a general framework that transforms a vision-language MLLM backbone into a unified model capable of joint comprehension and generation of sounding videos. The backbone LLM is flexible and model-agnostic, allowing for various architectures and scales. In the manuscript, we adopt Qwen2.5-VL-7B-Instruct [5] as our primary backbone, and this section details the architecture configuration. Investigation on alternative base LLMs is also provided in Sec. E.2.

**Backbone MLLM**. Qwen2.5-VL-7B follows a decoder-only architecture, consisting of 28 Transformer blocks with a hidden size of 3584. During training, the original model parameters are kept frozen. We inject trainable LoRA [27] modules into each linear layer, with hyperparameters set to $r = 128$ and $\alpha = 256$. After training, the LoRA weights can be merged back into the LLM via reparameterization, incurring no additional overhead during inference.

**Visual Encoder and Projector**. We adopt the ViT and Merger modules from Qwen2.5-VL [5] as the visual encoder and projector, respectively, and keep their parameters frozen during training. The ViT has a hidden size of 1280 and a patch size of 14. After feature extraction, the Merger applies a 2×2 patch grouping and projects it into the LLM token embedding space with a dimension of 3584.

**Audio Encoder and Projector**. We adopt BEATs [10] as the audio encoder and keep it frozen during training. A learnable 2-layer MLP is used as the projector to map the 768-dimensional features produced by BEATs to the 3584-dimensional token embedding space of the LLM. For all newly introduced MLPs in this work, the intermediate size is set as $4\times$ of the output embedding size, and we omit this detail in the subsequent descriptions.

**Audio-Video Synchronizer (SyncFusion)**. It consists of an MHSA layer with a hidden size of 3584 and 8 attention heads, with the FFN also implemented as a simple 2-layer MLP.

**Audio-Video Generator and Projector**. We use JavisDiT-v0.1 [39] as the DiT backbone, which consists of 28 layers of dual-stream DiT blocks with a hidden size of 1152. The text encoder is T5-XXL [59], with an embedding size of 4096 and a context length (*e.g.*, number of learnable semantic queries) of 300. The spatiotemporal prior encoder is ImageBind [24], with an embedding size of 1024 and a context length (*e.g.*, number of learnable spatiotemporal prior queries) of 77. Both the semantic projector and the spatiotemporal prior projector are 2-layer MLPs, mapping the 3584-dimensional LLM hidden states to 4096- and 1024-dimensional condition embeddings, respectively.

## B.2 Training Configuration

Building upon the strong visual understanding and basic instruction-following capabilities of the Qwen2.5-VL [5] backbone, we adopt the three-stage training strategy described in Sec. 4 to develop JavisGPT for joint understanding and generation of synchronized sounding videos. Table A1 summarizes the key configurations, and the training and data details are provided below. All models are trained on 8 NVIDIA A100-80GB GPUs.

**Stage I: Multimodal Pretraining.** This stage aims to equip the model with basic audio-video understanding and generation capabilities. (1) *Understanding*: We use 600K audio-text pairs to enhance the audio understanding ability. Tasks include audio captioning and question answering, with data sourced from AudioSet [23], AudioCaps [30], VGGSound [8], WavCaps [48], Clotho [19], ESC50 [54], GTZAN [66], MACS [47], and UrbanSound8K [61]. Training is conducted using next-token prediction loss. (2) *Generation*: We use 1.5M audio-video-caption triples from TAVG-Bench [46] to pretrain the generation component, using caption alignment loss $\mathcal{L}_{align}$ only. During this stage, the trainable components include the audio projector, query embeddings, and condition projector. We set the learning rate to 1e-3 and train for one epoch.

**Stage II: Audio-Video FineTuning.** This stage aims to enhance the model's ability to capture and maintain synchronization between audio and video in both understanding and generation tasks. (1) *Understanding*: We use 450K audio-video-text triplets from TAVGBench [46] and train on the sounding video captioning task using next-token prediction loss. (2) *Generation*: The same 450K triplets are reused for text-to-sounding-video generation, with a combination of caption loss and diffusion loss. In this stage, we additionally fine-tune the LoRA modules of the LLM, along with previously trainable components. The learning rate is set to 1e-4, and training runs for one epoch.

Table A1: **Detailed settings for progressive audio-video-synchronized training.**

| Setting | Stage-I | Stage-II | Stage-III |
|---|---|---|---|
| Purpose | MM-PreTrain | AV-FineTune | MM-InstTune |
| Tasks | A-Und + AV-Gen-Pre | AV-Und + AV-Gen | A/V/AV-Und + AV-Gen |
| Trainable Module | $\phi^a + Q^{c,s}, \phi^{c,s}$ | $\psi^{av} + Q^{c,s}, \phi^{c,s} + \Psi^{\text{LLM}}_{lora}$ | $\phi^a, \psi^{av} + Q^{c,s}, \phi^{c,s} + \Psi^{\text{LLM}}_{lora}$ |
| Trainable Params | 239.95M | 654.68M | 717.09M |
| Training Objective | $\mathcal{L}_{ntp} + \mathcal{L}_{align}$ | $\mathcal{L}_{ntp} + \mathcal{L}_{align} + \mathcal{L}_{diff}$ | $\mathcal{L}_{ntp} + \mathcal{L}_{align} + \mathcal{L}_{diff}$ |
| Training Samples | 600K + 1.5M | 450K + 450K | 600K |
| Training Epochs | 1 | 1 | 1 |
| Warm-up Epochs | 0.03 | 0.03 | 0.03 |
| Batch Size | 256 | 64 | 64 |
| Learning Rate | 1e-3 | 1e-4 | 1e-4 |
| Weight Decay | 0.0 | 0.0 | 0.0 |
| GPU Days (A100) | 6.5 | 14.2 | 9.3 |

**Stage III: Multimodal Instruction Tuning.** This stage is designed to equip the model with instruction-following and basic reasoning capabilities. (1) *Understanding*: We first construct the JavisInst-Und subset, a synchrony-aware audio-video QA dataset spanning three levels: entity, relation, and global comprehension. It consists of 110K samples derived and processed from TAVG-Bench [46]. Details can be found in Sec. C. To further enhance performance, we also incorporate 95K audio-video understanding samples from VideoLLaMA2 [14], including training splits from AVQA [89], MusicAVQA [41], and AVSD [3]. To prevent catastrophic forgetting, we additionally include 20K image understanding samples from LLaVA-OneVision [32], 60K video understanding samples from LLaVA-Video-178K [95], 550K audio comprehension samples from Stage I's dataset, and 20K audio-video caption samples from TAVGBench [46]. (2) *Generation*: Since no existing AV-instruction dataset is available for generation tasks, we construct the JavisInst-Gen subset based on repurposed data from TAVGBench. It covers a diverse set of tasks, including text-to-audio-video generation, in-context generation, and interleaved multi-turn conversation. More details are provided in Sec. C. Besides, 150K audio-video generation instances are also included to maintain JAVG capability learned from the previous stage.

### B.3 Evaluation Setup

In Sec. 6, we comprehensively evaluate the multimodal understanding and generation performance of JavisGPT, and here we introduce the detailed evaluation setups.

**Multimodal comprehension.** We follow the evaluation protocol of VideoLLaMA2 [14], and conduct unified benchmarking across visual understanding, audio understanding, and synchronized audio-video understanding for various multimodal large language models. The evaluation covers open-ended and multiple-choice formats, with single- or multi-turn dialogues. (1) *vi deo understanding* tasks are evaluated on three subsets: ActivityNet-QA [6], Perception Test [53], and MVBench [35]; (2) *audio understanding* is evaluated on Clotho-AQA [38] and TUT2017 [49]; and (3) *audio-video understanding* includes AVQA [89], MusicAVQA [41], and AVSD [3]. All models are evaluated in the zero-shot setting using greedy decoding to ensure reproducibility. Following VideoLLaMA2 [14], we report the average accuracy as the primary evaluation metric. For multiple-choice datasets, accuracy is calculated via lexical match, while open-ended responses are evaluated with the assistance of GPT-3.5 [1]. To ensure fair comparison, input videos are uniformly sampled to 16 frames, and audio is preserved at 16 kHz across all models.

**Multimodal generation.** For audio-video generation evaluation, we follow the protocol of Javis-DiT [39], using 1,000 text-to-audio-video samples from JavisBench-mini [39] for a comprehensive assessment. All models are evaluated with 4-second videos at 240P resolution and 24fps, and audio sampled at 16kHz.

A comprehensive evaluation of quality, consistency, and synchrony for audio-video generation results is provided. Here, we briefly introduce the mechanisms of each evaluation dimension:

Table A2: **Comparison on sounding video datasets for instruction tuning.**

| Dataset | Quantity | Diversity | AV-Synchrony | Multi-Turn | Reasoning |
|---|---|---|---|---|---|
| *Comprehension* | | | | | |
| MusicAVQA [41] | 32K | ✗ | ✗ | ✗ | ✗ |
| AVQA [89] | 40K | ✓ | ✗ | ✗ | ✗ |
| AVSD [3] | 8K | ✓ | ✗ | ✓ | ✗ |
| **JavisInst-Und** (Ours) | **110K** | ✓ | ✓ | ✓ | ✗ |
| *Generation* | | | | | |
| MosIT [81] | 5K | ✗ | ✗ | ✓ | ✓ |
| **JavisInst-Gen** (Ours) | **90K** | ✓ | ✓ | ✓ | ✓ |

- **Audio / Video Quality**: measuring the perceptual quality of the generated audio and video, including (1) *Fréchet Video Distance (FVD)*: $\text{FVD} = \|\mu_r - \mu_g\|_2^2 + \text{Tr}(\Sigma_r + \Sigma_g - 2(\Sigma_r \Sigma_g)^{1/2})$, where $(\mu_r, \Sigma_r)$ and $(\mu_g, \Sigma_g)$ are the mean and covariance of ground-truth and generated video features extracted by a pretrained I3D encoder [7]. Lower is better, indicating the generated video distribution is closer to the real one; (2) *Kernel Video Distance (KVD)*: similar to FVD, but estimates distribution differences via a kernel-based method (Kernel Inception Distance style), which is more stable on smaller datasets; lower is better; and (3) *Fréchet Audio Distance (FAD)*: same concept as FVD, but computed on audio features extracted by a pretrained AudioClip model [26], measuring distribution distance between generated and real audio; lower is better.

- **Text Consistency**: evaluating how well the generated audio and video semantically match the input text description, including (1) *ImageBind [24] text-video cosine similarity*: $\text{sim}(t, v) = \frac{f_{\text{text}}(t) \cdot f_{\text{video}}(v)}{\|f_{\text{text}}(t)\| \cdot \|f_{\text{video}}(v)\|}$; (2) *ImageBind text-audio cosine similarity:* same process but with the audio encoder $f_{\text{audio}}$; (3) *CLIP-Score*: using CLIP [56] to compute semantic similarity between text and video (video frames are sampled, encoded, and averaged); and (4) *CLAP-Score*: using CLAP [83] to compute semantic similarity between text and audio.

- **Audio–Video Semantic Consistency**: measuring the semantic alignment between generated audio and generated video, including (1) *ImageBind audio-video cosine similarity*, encoding both modalities into the same space and computing cosine similarity between video and audio features; and (2) *Audio-Visual Harmony Score (AVHScore)*: introduced in TAVGBench [46] as a way to quantify how well the generated audio and video align semantically in a shared embedding space. It is defined by computing the cosine similarity between each video frame and the entire audio, then averaging across all frames: $\text{AVHScore} = \frac{1}{N} \sum_{i=1}^{N} \cos(f_{\text{frame}}(v_i), f_{\text{audio}}(a))$. A higher AVHScore indicates stronger audio–video semantic consistency. Note that we remove the CAVP-Score [43] used in JavisDiT [39] because this metric keeps a range from 0.798 to 0.801 and cannot capture the difference when evaluating semantic consistency.

- **Audio–Video Spatio-Temporal Synchrony**: evaluating spatiotemporal alignment in generated audio-video pairs, focusing on *JavisScore*: a new metric proposed in JavisDiT [39]. The core idea is to use a sliding window along the temporal axis to split the audio-video pair into short segments. For each segment, compute cross-modal similarity with ImageBind and take the mean score: $\text{JavisScore} = \frac{1}{N} \sum_{i=1}^{N} \sigma(a_i, v_i), \quad \sigma(v_i, a_i) = \frac{1}{k} \sum_{j=1}^{k} \underset{\min}{\text{top-}k} \{\cos(E_v(v_{i,j}), E_a(a_i))\}$.

## C    JavisInst-Omni Dataset Details

### C.1    Motivation

To advance instruction tuning for multimodal large language models in sounding video scenarios, we introduce two new datasets: JavisInst-Und and JavisInst-Gen, which support open-ended QA for a wide range of understanding and generation tasks, respectively. This is motivated by our observation that existing datasets suffer from significant limitations in terms of scale, capability coverage, and task diversity. Tab. A2 summarizes the comparison with existing mainstream datasets. Below, we highlight the key differences and contributions of our proposed datasets.

**Audio-video comprehension.** While AVQA [89], MusicAVQA [41], and AVSD [3] provide some diversity or dialogue support, they mainly fall short in modeling and evaluating audio-video synchrony (AV-Synchrony). Specifically, MusicAVQA [41] is limited to musical scenarios and lacks content diversity. AVQA [89] covers a wide range of everyday scenes, but the majority of its questions are simple existential queries such as "what" or "where", with little emphasis on event-level synchrony between audio and video, and it does not support multi-turn interactions. AVSD [89] partially addresses the multi-turn dialogue limitation, but still does not emphasize AV-synchrony, and its relatively small scale (8K samples) makes it unsuitable for large-scale training. To address these limitations, we introduce JavisInst-Und, which not only inherits the strengths of prior datasets but also emphasizes on synchrony-aware comprehension and expands the scale to 110K samples, providing a solid foundation for large-scale synchronized audio-video understanding.

**Audio-video generation.** In fact, there currently exists no instruction dataset specifically designed for synchronized audio-video generation. Although the MosIT dataset (see Tab. A2) includes multi-turn dialogues and reasoning capabilities, it does not contain direct training samples for generating audio-video content. To fill this gap, we introduce JavisInst-Gen, a dataset of 90K samples covering a wide range of everyday conversational scenarios. It supports synchronized AV generation with multi-turn interaction and reasoning capabilities, addressing the current limitations in the field.

JavisInst-Und and JavisInst-Gen are primarily constructed from TAVGBench[46] and InstV2V[13]. We leverage GPT-4o [1] and video-to-audio tools [94] for QA generation and data processing, and Fig. A3 showcases some representative examples for multi-turn comprehension and generation.

## C.2 JavisInst-Und

**Taxonomy**. As detailed in Tab. A3, we construct a hierarchical taxonomy that captures information from local to global levels, covering 3 major categories and 10 subcategories: (1) *Entity-level*, including existence, alignment, grounding, and counting tasks, aiming to capture fine-grained auditory and visual characteristics of each individual sounding event. (2) *Relation-level*, includes spatial, temporal, and causal relationships in modeling the spatiotemporal and causal interactions between different sounding events. and (3) *Global-level*, including theme, emotion, and atmosphere analysis, aiming to capture the overall emotion and thematic expression of audio-video content.

**Construction**. We propose an efficient method for constructing JavisInst-Und by reusing audio-video samples from AudioSet [23] and leveraging the corresponding text captions provided in TAVGBench [46]. With the powerful language capabilities of GPT-4o [1], we generate diverse QA pairs for audio-video understanding.

• *Single-turn audio-video understanding*: For each of the 10 predefined dimensions, we design prompt templates (as exemplified in Fig. A1) and adopt a few-shot prompting strategy to guide GPT-4o in generating category-specific QA pairs based on the given audio-video caption. We randomly generate QA exercises from 3-4 categories for each sounding video caption.

• *Multi-turn audio-video understanding*: For videos from the same source in TAVGBench, we construct multi-turn dialogues in two ways: (1) randomly compositing multiple single-turn QA samples into a multi-turn session ("Composite"); and (2) first curating a single-turn instruction-based audio-video generation dialogue (see Sec. C.3), and then augmenting it with the corresponding single-turn QA samples above to create 2–3 turn "Gen2Und" conversations, further improving scene diversity.

In total, we construct a dataset of 110K QA samples in JavisInst-Und (see Fig. 5 for distribution), supporting a broad range of audio-video understanding tasks and applications.

## C.3 JavisInst-Gen

**Taxonomy**. Table A4 provides a detailed overview of the three types of audio-video generation tasks included in our dataset, along with representative examples: (1) *Instruction-based Generation*. This is the core text-to-audio-video generation task. We distinguish between formal-style (more written and detailed) and colloquial-style (more casual and concise) instructions to better reflect real-world dialogue scenarios. (2) *Conditional Generation*. This task aims to generate new audio-video content based on modalities beyond text (*e.g.*, images, videos, audio, or any combination thereof) as a context condition. It broadens application potential and includes tasks such as audio-video extension for long-

Table A3: **Clarification of the category taxonomy of JavisInst-Und**.

| Aspect | Category | Description | Example (explanation omitted) |
|---|---|---|---|
| **Entity** | Existence | Whether an object with a specific audio characteristic appears in the video. | *Que*: Is the cat meowing? *Ans*: No. |
| | Alignment | Whether the object making a certain sound and the object in the video are the same one. | *Que*: Does the lion in the video make the roaring sound in the audio? *Ans*: Yes. |
| | Grounding | The position of the object with a certain audio characteristic in the video. | *Que*: Where is the dog that is barking? *Ans*: Near a red car. |
| | Counting | The number of some objects in the audio and video. | *Que*: How many people can be identified in the video and audio? *Ans*: Two. |
| **Relation** | Spatial | The relative position of the different objects that make the sounds. | *Que*: Where is the barking dog in relation to the cat that is meowing? *Ans*: Over the cat. |
| | Temporal | The temporal relationship between the audio and video. | *Que*: Does the honking sound appear before the pedestrian crosses the street? *Ans*: Yes. |
| | Causal | The activity relationships between different objects. | *Que*: What causes the loud crash sound? *Ans*: The child dropping a glass. |
| **Global** | Emotion | The emotions expressed by the characters appearing in the video. | *Que*: What emotion is the woman expressing? *Ans*: Happy. |
| | Atmosphere | The common atmosphere in the video and audio. | *Que*: How about the atmosphere shown in the video and audio? *Ans*: Chaotic. |
| | Theme | Whether an object with a specific audio characteristic appears in the video. | *Que*: What is the overall theme of the video? *Ans*: Family picnic. |
| **MultiTurn** | Composite | 2-3 rounds of conversations about the same sounding video. | *Que*: Where is the source of the machine gun firing sound? *Ans*: From the helicopter. *Que*: When does the sound of the machine gun firing occur? *Ans*: While the helicopter is flying. |
| | Gen2Und | Model first generates a sounding video, then answers a corresponding question. | *Que*: Generate a video that follows these guidelines: as the boat tow truck pulls ... *Ans*: Sure! *Que*: Does the boat being lifted out of the water produce any sound in the audio? *Ans*: No. |

form video generation and coarse-grained AV editing. (3) *Multi-turn Conversation*. This includes more complex task setups involving proactive reasoning and understand-then-generate (Und2Gen) scenarios, where the model must first comprehend prior context before generating. These tasks further enrich the diversity and complexity of the dataset.

**Construction**. JavisInst-Gen is also efficiently constructed by organizing data around the three defined types of audio-video generation tasks:

• *Instruction-based Generation*: This is the core task, where the model is required to generate synchronized audio-video content conditioned on a given text caption. To support this, we leverage GPT-4o [1] to construct 3,000 diversified prompt templates, and randomly sample 45K audio-video captions from TAVGBench [46] to combine with these templates, forming a wide range of instruction-based generation instances. To enhance linguistic diversity, approximately 20% of the instructions are further paraphrased by GPT-4o-mini (for cost efficiency), making them more aligned with natural, conversational instructions typically found in real-world applications.

• *Conditional Generation*: This task follows a similar construction strategy to instruction-based generation, where we synthesize a large number of conditional generation cases by randomly combining prompt templates with existing audio-video captions, followed by modality-specific input processing. For example, in A2V tasks, the audio is extracted as input; and in I2AV, the first frame of the video is used as input. For AV-Extension, we sample two overlapping clips (*e.g.*, 1–2s overlap) from a $\tilde{1}0$s video in TAVGBench, using the first as input and the second as the generation target. As for

Figure A1: **Prompt for spatial relationship question-answering data synthesis.** Here `{caption}` is the placeholder for a given audio-video caption.

Figure A2: **Prompt for proactive audio-video generation data synthesis.**

AV-Edit, since TAVGBench (and other related datasets) lack suitable editing instructions, we adopt an alternative approach: we randomly sample 5K video editing pairs from InstV2V[13], and apply

Table A4: **Clarification of the category taxonomy of JavisInst-Gen**.

| Task | Category | Description | Example (response omitted) |
|------|----------|-------------|----------------------------|
| **Instruct** | Formal | Sounding video generation with formal instruction. | *Que*: Compose a thorough video piece featuring narration and imagery. The sound of a motorcycle engine ... |
| | Colloquial | Sounding video generation with colloquial instruction. | *Que*: Make this into a video right away. A female vocalist is performing ... |
| **Condition** | V2A | Generate an audio for input video. | *Que*: Mount an audio track to the visuals. The intense and fast-paced ... |
| | A2V | Generate a video for input audio. | *Que*: Produce a video narrative: a robotic vacuum cleaner is parked ... |
| | AI2V | Animate an image with input audio. | *Que*: Make a video with the picture and soundtrack. A motorcycle revving ... |
| | I2AV | Animate an image and make an audio. | *Que*: Compose a video from this image, with sound. As the toy police car sits ... |
| | AV-Ext | Temporally extend the sounding video. | *Que*: Continue from the stopping point. A man wearing a pink t-shirt ... |
| | AV-Edit | Edit the sounding video as supposed. | *Que*: Make the concept art look like ancient Egyptian hieroglyphs. |
| **MultiTurn** | Proactive | Model first extracts user preference, then generates a corresponding sounding video. | *Que*: What do you think about oranges as a fruit? *Ans*: They're really invigorating and full of flavor! *Que*: Create a video where a car driving ... with a fruit balanced on top of it. |
| | Rethink | Model first supplements a detailed request, then generates a corresponding sounding video. | *Que*: Create a video where a person crumples paper and ... *Ans*: Do you want the sound of paper being crumpled up can be heard as ...? *Que*: Yes! |
| | Und2Gen | Model first answers an audio-video-related question, and the generate/extend a corresponding sounding video. | *Que*: What emotion is the crowd expressing? *Ans*: Surprise and a mixture of applause. *Que*: Show me a video of a crowd with silence and sacredness. |

FoleyCrafter[94] to generate corresponding audios for both the pre-edit and post-edit video clips, thus enriching training scenarios for AV-based editing tasks.

• *Multi-turn Conversation*: We carefully design three representative and practical forms of multi-turn generation: proactive generation, rethink generation, and understanding-to-generation (Und2Gen). In proactive generation, the model is expected to identify user preferences from prior dialogue and incorporate them into the generated sounding video. In rethink, the model receives an ambiguous initial instruction and proactively suggests a more complete and specific generation prompt.

To construct such data, we reuse sounding videos from TAVGBench as generation targets and employ GPT-4o to synthesize dialogues based on their corresponding captions. Specifically: (1) for proactive scenarios, we remove one key entity or concept from the caption and create a preceding dialogue turn that embeds a user preference, simulating preference-aware generation (see prompt template in Fig. A2). (2) for rethink, we generate a highly abstract version of the caption as the user's vague instruction. Then, the original detailed caption is reformatted using instruction generation templates to serve as the model's suggested clarification.

As for Und2Gen, the model first receives a sounding video and engages in one round of understanding dialogue (from Sec. C.2), followed by a generation task such as producing a new audio-video clip or performing AV-extension based on prior understanding. Inspired by NExT-GPT [81], we use BERTScore [92] to identify semantically similar caption pairs in TAVGBench, and instruct GPT-4o to construct conditional generation prompts accordingly.

In total, JavisInst-Gen contains 90K high-quality QA samples (see detailed statistics in Fig. 5), supporting a wide range of real-world generation scenarios.

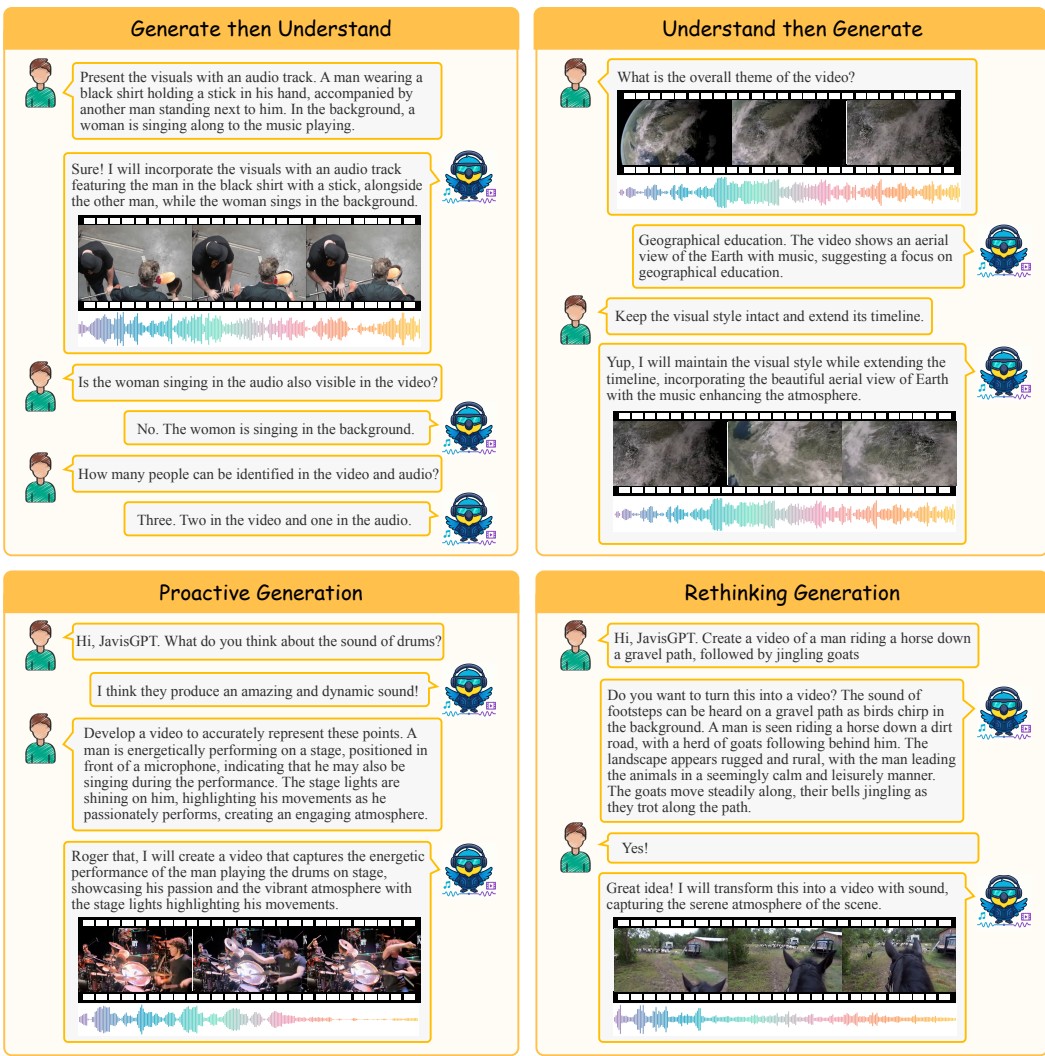

Figure A3: **Examples of multi-turn comprehension and generation from JavisInst-Omni.**

# D Architecture of Joint Comprehension and Generation of Sounding Videos

## D.1 Design Choices of Generative MLLM

Designing a unified architecture for both comprehension and generation tasks remains a central challenge. As there are no previous exploration in JAV domain (we are the first one), we mainly summarize and compare three representative paradigms in Fig. A4 under the text-image context:

**LLM+DiT**. Our JavisGPT adopts a modular yet effective LLM + DiT architecture, which has been widely verified in recent multimodal works [81, 51] for its balance between efficiency and generation quality. This architecture separates semantic understanding (handled by the LLM) from generative decoding (handled by a diffusion transformer), allowing task specialization and easier optimization. Moreover, the use of learnable queries and a conditioned DiT enables flexible generation from various modalities with strong temporal coherence.

**MonoLLMs**. Recent works [96, 84, 45] attempt to unify textual autoregression and multimodal diffusion within a single large language model. However, these MonoLLMs suffer from severe optimization difficulties due to conflicting training objectives and heterogeneous output formats [9]. The lack of architectural decoupling often leads to poor convergence and unstable generation, especially for high-dimensional outputs like audio or video.

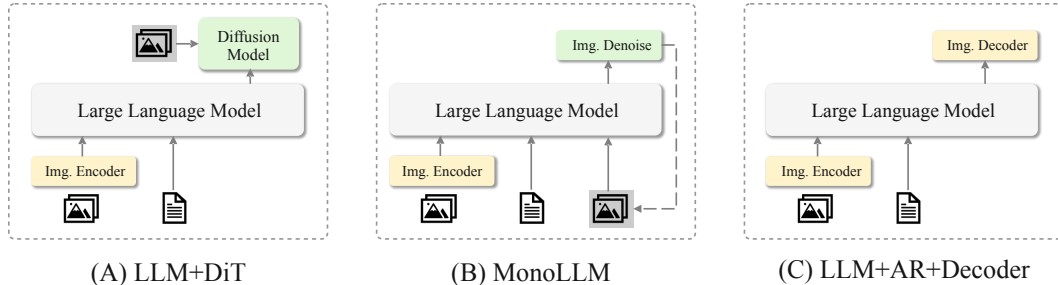

(A) LLM+DiT     (B) MonoLLM     (C) LLM+AR+Decoder

Figure A4: **Choices of unified architecture for joint understanding and generation. (A)** Our JavisGPT adopts the widely verified LLM+DiT architecture [81, 51] for efficiency and efficacy. **(B)** MonoLLMs [96, 84] that integrate textual auto-regression and multimodal diffusion into a single LLM, causing severe optimization difficulty. **(C)** The LLM+AR+Decoder paradigm [77, 82] brings extra training efforts on encoder-decoder modules for JAVG and high inference cost to predict numerous multimodal tokens in the auto-regression behavior.

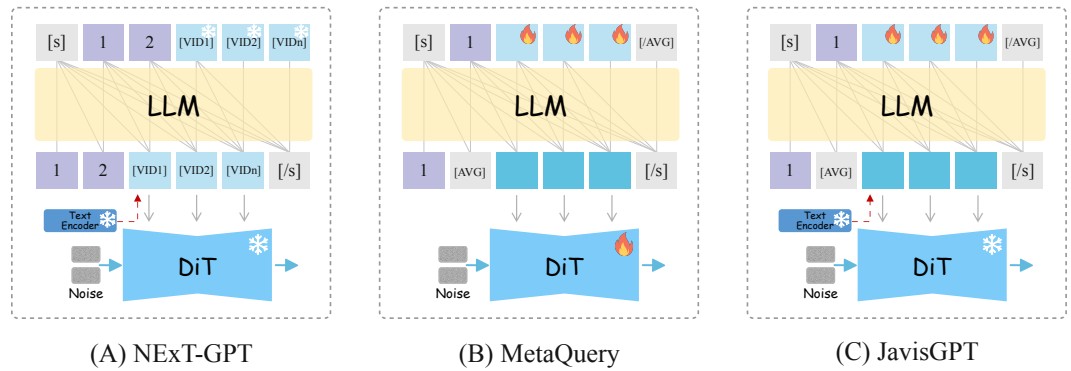

(A) NExT-GPT     (B) MetaQuery     (C) JavisGPT

Figure A5: **Choices for MLLM + DiT combination. (A)** NExT-GPT [81] suffers from degraded generation quality and diversity due to its forced regression of all generation instructions to a fixed set of special tokens (*i.e.*, <VID1>, <VID2>, · · · , <VIDn>). **(B)** MetaQuery [51] removes the support of a text encoder, which leads to training instability. **(C)** Our JavisGPT combines learnable query embeddings with a supportive text encoder, enabling stable training and the generation of high-quality and diverse audio-video content.

**LLM+AR+Decoder**. The autoregressive encoder-decoder paradigm [77, 82, 75] introduces additional training complexity, which requires maintaining a heavy encoder-decoder pipeline to bridge multimodal embeddings with autoregressive prediction on discrete [82] or continuous [75] token embeddings and trying to avoid information loss. Besides, this design also incurs significant inference costs, as it must sequentially predict large volumes of multimodal tokens. For instance, when using 512 tokens per frame, generating a one-minute video clip (even without audio) at 24 fps would require generating $512 \times 24 \times 60 = 737{,}280$ tokens, making it less practical for real-time or high-resolution generation.

In conclusion, the LLM+DiT paradigm is the best option for current JAVG field, which has also been verified by our comprehensive experiments in Sec. 6. However, the LLM+AR+Decoder architecture may show potentials to further bridge the unified task of comprehension and generation. The key insight is simple and straightforward: after unifying understanding and generation AV representations into the same space, the generation embeddings can be also "seen" by comprehension tasks. As discussed in Sec. A, this feature can mutually strengthen both comprehension and generation capabilities after scaling up model capacity and data quantity. We hope this can bring more inspiration in the community.

## D.2    Design Choices of LLM-DiT Combination

In Fig. A5, we also compare several designs of LLM+DiT combination, which further highlights the advance of our JavisGPT's architecture in performance, diversity, and training stability.

**NExT-GPT**. NExT-GPT [81] formulates generation as a classification task over a fixed vocabulary of video IDs (*e.g.*, <VID1>, <VID2>, $\cdots$, <VIDn>), forcing all instruction prompts to regress to this small set of special tokens. This rigid formulation limits the expressiveness of conditional instructions and severely constrains the diversity and fidelity of generated audio-video outputs, which is consistent to NExT-GPT's poor generation performance in Sec. 6.1. Additionally, it restricts generalization to novel or fine-grained instructions outside the pre-defined token set.

**MetaQuery**. MetaQuery [51] introduces a learnable query mechanism but removes the auxiliary support of the text encoder of DiT, relying solely on query tokens to interface with the generation model. This approach often leads to training instability, especially when scaling to complex multimodal data like audio-video. The absence of semantically grounded text features weakens the model's ability to align instructions with generated content.

**JavisGPT**. Our JavisGPT integrates the advantages of both sides by combining learnable query embeddings with the semantic guidance of a pretrained text encoder. The text encoder offers strong grounding for conditional semantics, while the queries serve as adaptive, trainable intermediaries to control the generative diffusion model. This hybrid design results in stable training dynamics and supports the generation of high-quality, diverse, and instruction-aligned sounding videos. Moreover, it enables better generalization to a wide range of instructions, including free-form, multi-turn, or mm-conditional prompts.

# E Supplementary Investigation on JavisGPT

## E.1 Details for Human Evaluation on Interleaved Conversation

In Sec. 6.1, we present a comparative evaluation with UnifiedIO-2 [42] and NExT-GPT [81] in interleaved comprehension and generation scenarios. Since no existing benchmark is available for this setting, we conduct a human evaluation, as detailed below:

**Data Collection.** We use GPT-4o to construct 100 multi-turn dialogue samples based on the four scenarios illustrated in Fig. A3, including Gen2Und, Und2Gen, Proactive, and Rethink (25 samples each).

**Evaluation Criteria.** Three unbiased human annotators independently rated the model outputs on a scale of 0–5 across five dimensions:

1. *Instruction Following*: whether the model follows the user instruction to generate a coherent textual response and a matching audio-video output;
2. *Question Answering*: whether the model accurately understands the input audio-video content and provides correct answers;
3. *Generation Quality*: the overall quality, coherence, and synchrony of the generated audio-video content;
4. *Context Reasoning*: whether the model can integrate user preferences implied in the dialogue history into its response or generation;
5. *Proactive Thinking*: whether the model can recognize vague or under-specified instructions and proactively ask clarifying questions or complete the intent.

**Result Analysis.** The evaluation results are shown in Fig. 6, from which we draw several key conclusions:

- NExT-GPT shows poor instruction-following ability and often fails to generate audio-video outputs—only 33 out of 100 cases succeeded. This can be attributed to insufficient fine-tuning (only 5K samples used).
- Both NExT-GPT and UnifiedIO-2 suffer from low generation quality. As discussed in Sec. D.2, NExT-GPT's reliance on fixed special tokens for all instructions limits both generation diversity and fidelity. UnifiedIO-2, lacking a dedicated DiT-based generator, resorts to generating keyframes and stitching them into video, resulting in subpar temporal coherence and realism.
- JavisGPT significantly outperforms both baselines in question-answering, benefiting from the scale and diversity of the JavisInst-Omni dataset.
- Due to the lack of training on preference reasoning and proactive dialogue, both NExT-GPT and UnifiedIO-2 exhibit weak context reasoning and proactive generation, whereas JavisGPT con-

sistently shows stronger contextual understanding and responsiveness, underscoring its practical utility and superior alignment with real-world conversational needs.

The results further highlight the strong capability of our JavisGPT.

### E.2 Investigation on Backbone LLM

In the manuscript, we mainly take the Qwen2.5-VL [5] as the backbone LLM to build our JavisGPT. Here we provide a further investigation on the base LLM with a slightly weaker Qwen2-VL [76] model. After the same training data and procedure, we compare the performance variation in Fig. A6 for further analysis. Accordingly, upgrading the language backbone from Qwen2-VL to Qwen2.5-VL consistently improves performance across all evaluation tracks, including unimodal understanding (V-Und, A-Und), audio-visual comprehension (AV-Und), and audio-visual generation (AV-Gen). Notably, the gain is most pronounced in the generation task (AV-Gen), where Qwen2.5-VL outperforms Qwen2-VL by a substantial margin. These results highlight that a more capable language foundation model can significantly benefit not only text-based reasoning but also cross-modal alignment and response generation. The consistent improvements across both understanding and generation tasks validate the critical role of the LLM backbone in scaling multimodal capabilities of our JavisGPT.

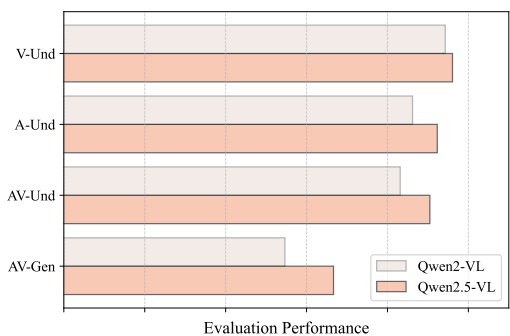

Figure A6: **Ablation on Backbone LLMs**.

## F  Qualitative Case Studies of JavisGPT

More understanding and generation cases can be found in Fig. A7.

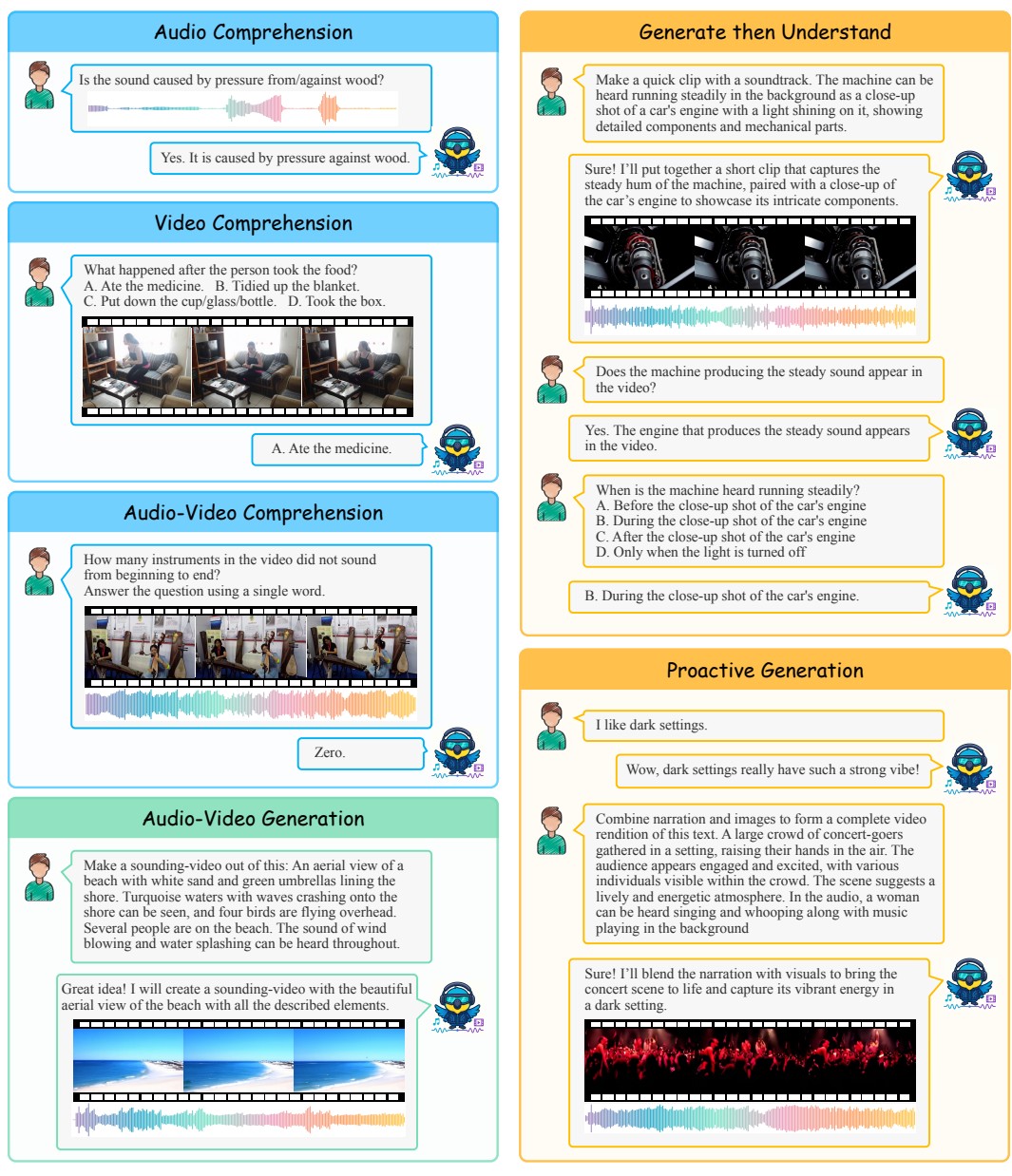

Figure A7: **More qualitative results for joint understanding and generation.**

