# OpenReview forum: "JavisGPT: A Unified Multi-modal LLM for Sounding-Video Comprehension and Generation"
_NeurIPS.cc/2025/Conference — NeurIPS 2025 spotlight_

### Official Review · Reviewer_SniZ · 2025-07-02

**Clarity:** 3
**Significance:** 3
**Originality:** 3
**Rating:** 4
**Confidence:** 4

**Summary:**

This paper constructs JavisGPT, a unified MLLM for joint audio-visual understanding and generation by cascading audio-visual LLM with JavisDiT. The paper includes training recipes for different stages and provides JavisInst-Omni, an audio-visual text dialogue dataset of over 200,000 samples synthesized by GPT-4o. Experiments show that JavisGPT outperforms existing omni models on audio-visual understanding and generation benchmarks, especially in complex temporal synchronization scenarios.

**Questions:**

* Can the JavisGPT support multi-turn audio-visual understanding and generation through specific data and architectural designs?
* Does the cascading architecture lead to bottlenecks in audio-visual generation? Are there any defects in efficiency?

**Ethical Concerns:**

["Major Concern: Improper research involving human subjects", "Major Concern: Data privacy, copyright, and consent"]

**Final Justification:**

The authors have comprehensively addressed all my questions. However, architecturally, I believe an end-to-end architecture might be more meaningful to the community. Therefore, I maintain my current positive score.

**Limitations:**

Yes

**Quality:**

3

**Strengths And Weaknesses:**

### Strengths

* Constructing a joint audio-visual understanding and generation system that achieves promising results on corresponding benchmarks is an extremely challenging yet profoundly significant task. It requires complex system design and overall optimization, which I deem highly non-trivial. Thus, the paper makes solid contributions.
* The authors' SyncFusion design alleviates the audio-visual asynchrony issue in previous dual-branch audio-visual methods, demonstrating practical value.
* The paper describes the architectural design, training schemes, relevant data (e.g., JavisInst-Omni), and implementation code, which positively promotes the development of the entire field.

### Weaknesses

* Strictly speaking, the proposed model is not the first joint audio-visual understanding and generation model—similar architectures and concepts have been presented in AV2AV [1] and MultiDialog [2], which diminishes the work's originality.
* The authors primarily focus on single-turn "video-audio" pair understanding and generation. Exploring multi-turn "video-audio" scenarios would significantly enhance the work's significance.
* The model employs a cascading approach with an external DiT rather than an end-to-end unified architecture for constructing the joint understanding-generation model, which diminishes part of the work's significance.

[1] AV2AV: Direct Audio-Visual Speech to Audio-Visual Speech Translation with Unified Audio-Visual Speech Representation
[2] Let’s Go Real Talk: Spoken Dialogue Model for Face-to-Face Conversation

---

> ### Author Rebuttal · Authors · 2025-07-31
>
> > **W1**: Strictly speaking, the proposed model is not the first joint audio-visual understanding and generation model—similar architectures and concepts have been presented in AV2AV and MultiDialog, which diminishes the work's originality.
>
>
> **Ans**: We thank the reviewer for this comment. While AV2AV, MultiDialog, and the proposed JavisGPT all involve audio-visual modalities, our originality lies in the following aspects:
>
> 1. **Conceptual motivation**: AV2AV and MultiDialog are limited to the specific task of talking-head generation with speech synthesis. In comparison, JavisGPT extends the paradigm to general natural audio-visual understanding and generation, which is more comprehensive and challenging, and offers broad application potential.
> 2. **Architecture**: AV2AV relies on a Transformer encoder-decoder for style transfer, while MultiDialog adopts an autoregressive language-speech model for conversation but completely discards textual input and output. Only JavisGPT employs the more mainstream encoder-LLM-decoder architecture, supporting free-form inputs across text, audio, and visual modalities, and enabling both textual dialogue and audio-visual generation. This design offers superior extensibility and generalization capability.
>
> We will incorporate this discussion into the revised manuscript to better highlight our originality and novelty.
>
> > **W2**: The authors primarily focus on single-turn "video-audio" pair understanding and generation. Exploring multi-turn "video-audio" scenarios would significantly enhance the work's significance.
>
> > **Q1**: Can the JavisGPT support multi-turn audio-visual understanding and generation through specific data and architectural designs?
>
> **Ans**: We would like to clarify that JavisGPT is inherently capable of multi-turn audio-visual understanding and generation. Architecturally, this is enabled by formatting dialogue history into structured prompt templates (e.g., `<user>...</user><assistant>...</assistant>` repeated over turns), which allows the model to retain and use context. Examples of multi-turn interactions are shown in Fig. 1 and Fig. 9.
>
> In our training set, we incorporated a wide range of multi-turn audio-visual scenarios into the training corpus JavisInst-Omni (Fig. 5), including both understanding and generation tasks. Furthermore, Fig. 6 reports human evaluation results conducted on interleaved multi-turn dialogues, further validating the the effectiveness of JavisGPT beyond single-turn settings.
>
> > **W3**: The model employs a cascading approach with an external DiT rather than an end-to-end unified architecture for constructing the joint understanding-generation model, which diminishes part of the work's significance.
>
> > **Q2**: Does the cascading architecture lead to bottlenecks in audio-visual generation? Are there any defects in efficiency?
>
> **Ans**: We thank the reviewer for raising this important point, which is indeed a critical consideration, and we also have discussed it in detail in Section 6 and Appendix A.1 of the paper. Below, we summarize the key arguments:
>
> 1. **Effectiveness**: Under current technological constraints, the LLM+DiT framework adopted by JavisGPT is the most robust and high-performing approach[1-2]. A fully end-to-end, autoregressive multimodal architecture would require mature multimodal tokenizers, extremely large-scale training data, and highly stable training infrastructure[3-4]. Those conditions are not yet sufficiently mature in the audio-visual field. In practice, such an end-to-end model may underperform compared to our cascaded framework.
>
> 2. **Efficiency**: Contrary to potential concerns, the current cascaded design is in fact efficient. As audio-visual tokens are highly dense and numerous[3-4], a fully autoregressive, token-by-token generation process would severely bottleneck inference speed, which is well-documented in works such as SelfTok[4]. In comparison, JavisGPT’s use of a query-based interface between the LLM and DiT—similar to MetaQuery[2], enabling more efficient and scalable generation.
>
> 3. **Architectural perspective**: From an LLM-centric viewpoint, the DiT can be seen as a large audio-visual decoder (analogous to a text detokenizer). Even in a fully autoregressive setup, a DiT-like component would likely still be necessary to maintain generation quality[3-5]. Therefore, we believe the current JavisGPT framework remains both practical and conceptually insightful for future developments.
>
> ---
>
> [1] Wu S, Fei H, Qu L, et al. Next-gpt: Any-to-any multimodal llm[C]//Forty-first International Conference on Machine Learning. 2024.
>
> [2] Pan X, Shukla S N, Singh A, et al. Transfer between modalities with metaqueries[J]. arXiv preprint arXiv:2504.06256, 2025.
>
> [3] Tong S, Fan D, Zhu J, et al. Metamorph: Multimodal understanding and generation via instruction tuning[J]. arXiv preprint arXiv:2412.14164, 2024.
>
> [4] Wang B, Yue Z, Zhang F, et al. Discrete visual tokens of autoregression, by diffusion, and for reasoning[J]. arXiv e-prints, 2025: arXiv: 2505.07538.
>
> [5] Open AI. Introducing 4o image generation. https://openai.com/index/
> introducing-4o-image-generation/, 2025.
>
> ---
>
> ## Ethical Statement
>
> Reviewer SniZ has flagged our submission with Major Ethical Concern on both human and data issues:
>
> > Major Concern: Improper research involving human subjects, Major Concern: Data privacy, copyright, and consent
>
> We hope to clarify that:
> 1. The human evaluation study involved three volunteers providing preference judgments. No sensitive personal data was collected. All participants gave informed consent, and institutional ethics approval was not required under our institution's policy.
> 2. The proposed JavisInst-Omni dataset is constructed entirely from publicly available data sources. It does not involve any private or proprietary content, human annotation, or personal information. There are no concerns regarding data privacy, copyright, or consent.

---

> > ### Comment · Reviewer_SniZ · 2025-08-03
> >
> > I thank the authors for providing a detailed and comprehensive rebuttal.
> >
> > The authors have provided a comprehensive response to the three points I questioned:
> >
> > 1. They addressed my concern regarding the comparison between JavisGPT and similar works such as AV2AV and MultiDialog.
> > 2. They resolved my doubt about whether JavisGPT is capable of multi-turn dialogues.
> > 3. They explained why a cascading architecture needs to be adopted.
> >
> > Therefore, I still maintain my positive score.

---

> > > ### Author Response · Authors · 2025-08-05
> > >
> > > Thank you for your positive response and continued support. We truly appreciate your insightful comments, which helped us improve the clarity and quality of our paper.
> > >
> > > Best  regards,
> > >
> > > Authors

---

### Official Review · Reviewer_3pH1 · 2025-07-02

**Clarity:** 3
**Significance:** 2
**Originality:** 2
**Rating:** 4
**Confidence:** 3

**Summary:**

The paper present JavisGPT, an encoder‑LLM‑decoder system that unifies synchronized audio‑video (AV) comprehension and generation. Key ingredients include:

(1) SyncFusion: a cross‑attention module that merges frame‑aligned audio features into video patches, yielding SyncAV tokens for fine‑grained spatio‑temporal reasoning.

(2) Hierarchical queries (JavisQuery / JavisCond / ST‑prior) that map LLM hidden states to a frozen JAV‑DiT diffusion generator, enabling one‑shot, temporally coherent AV synthesis.

(3) A three‑stage training pipeline—multimodal pre‑train, AV fine‑tune, and instruction tuning—driven by the 200 k‑dialogue JavisInst‑Omni dataset .

**Questions:**

1. How does SyncFusion differ fundamentally from earlier audio–video fusion layers such as the Q-Former/FiLM approaches used in AVFormer and NExT-GPT? Where is the genuine innovation beyond frame-aligned cross-attention?

2. If the temporal-spatial prior query is removed—keeping only JavisQuery and JavisCond—how much do synchrony and generation quality drop? Please provide an ablation isolating its contribution.

3. Could the frozen DiT be replaced with a lighter autoregressive or VQ-GAN decoder without retraining the LLM adapters? Any preliminary experiments or rationale?

**Ethical Concerns:**

["NO or VERY MINOR ethics concerns only"]

**Final Justification:**

My concerns has been solved. Although the novelty of this paper is limited, I think this is a good paper.

**Limitations:**

Yes

**Quality:**

2

**Strengths And Weaknesses:**

**Strengths**

(1) Covers eight comprehension benchmarks, text‑to‑AV generation metrics, and a human study on interleaved dialogues; baseline list is current and competitive.

(2) Build an model that simultaneously understands and generates fully synchronized video and audio with a shared backbone.


**Weaknesses**

(1) Limited conceptual novelty. Most of JavisGPT’s building blocks already exist in the literature and the paper does not clearly articulate what is fundamentally new beyond their specific combination:  SyncFusion is essentially a Q‑Former/FiLM‑style cross‑attention layer applied to frame‑aligned audio features; similar audio‑video fusion schemes appear in AVFormer [1]. Hierarchical queries (JavisQuery / JavisCond / ST‑prior) resemble the two‑level adapter hierarchy of Control‑Net and Uni‑Adapter, while the frozen DiT decoder mirrors Make‑A‑Video and VideoComposer. The three‑stage training recipe (multimodal pre‑train → task fine‑tune → instruction tuning) is standard practice inherited from LLaVA, MiniGPT‑4, and NExT‑GPT.

(2) The new 200 k “JavisInst‑Omni” dialogues are synthetically generated from GPT‑4 with task‑specific prompts, following LLaVA‑v1.5 and NExT‑GPT‑Lite, rather than collected from humans.

[1] Seo P H, Nagrani A, Schmid C. Avformer: Injecting vision into frozen speech models for zero-shot av-asr[C]//Proceedings of the IEEE/CVF Conference on Computer Vision and Pattern Recognition. 2023: 22922-22931.

---

> ### Author Rebuttal · Authors · 2025-07-31
>
> > **W1**:  Limited conceptual novelty. Most of JavisGPT’s building blocks already exist in the literature and the paper does not clearly articulate what is fundamentally new beyond their specific combination.
>
> **Ans**: We appreciate the reviewer’s concern regarding conceptual novelty. As discussed in Sec.4 Contribution Statement in this paper, we agree that most individual components may be known in literature, but we believe the following task-specific adaptations and system-level contributions are non-trivial:
>
> - **Unified system for audio-video understanding and generation:**
> JavisGPT is the first framework that jointly handles instruction-driven and generalized audio-visual understanding and generation within a multi-turn dialogue setup, with a shared backbone and efficient training pipeline.
>
> - **Empirical validation and design insights:**
> All key components are supported by systematic experiments and ablations, which also provides some insightful Findings. For example, we observe that comprehension task improves generation quality, but not vice versa. We also discuss the architectural implications of this asymmetry, offering perspectives that may inform future model design in Appendix A.1.
>
> - **[Minor] Some components are not exactly the same as existing works**:
> Modules such as SyncFusion and hierarchical queries are not direct copies of prior designs, but adapted to solve AV-specific challenges, i.e., capturing and maintaining the frame-level audio-video synchronization (details refer to our response to Q1).
>
> As noted by the other reviewers, the system design, model implementation, dataset curation, and emripical findings provide **practical value and meaningful inspiration for future research** in the field.
>
> > **W2**:  The new 200 k “JavisInst‑Omni” dialogues are synthetically generated from GPT‑4 with task‑specific prompts, following LLaVA‑v1.5 and NExT‑GPT‑Lite, rather than collected from humans.
>
> **Ans**: We acknowledge the reviewer’s concern and also view this as the potiential limitation. However, while our JavisInst-Omni data is generated via GPT-4o, we argue that well-designed synthetic data can provide scalable and diverse supervision, especially for multimodal tasks where annotation is costly. This approach has proven effective in ShareGPT4Video[1], and our results show that it significantly improves instruction-following ability across modalities.
> In the future, we plan to incorporate human annotations (such as preference data) to further enhance both the fidelity and diversity of the dataset.
>
>
> > **Q1**: How does SyncFusion differ fundamentally from earlier audio–video fusion layers such as the Q-Former/FiLM approaches used in AVFormer and NExT-GPT? Where is the genuine innovation beyond frame-aligned cross-attention?
>
> **Ans**: We hope to clarify the insights behind SyncFusion as follows:
>
> 1. **Structural difference from Q-Former**: Q-Former uses learnable query tokens to extract and compress information from audio and video modalities. This process often introduces *training difficulty and information loss*[2], which can hinder effective audio-video synchronization. In comparison, SyncFusion directly injects audio information into existing visual tokens, enabling simple yet effective frame-level alignment, as supported by the quantitative results in Table 4.
>
> 2. **Insights beyond frame-aligned cross-attention**: As illustrated in Figure 3, SyncFusion enables the construction of a *new modality*, where the resulting *SyncAV token* $e_{i,j}^{t} \in \mathbb{R}^{C}$ represents a sounding event localized at the $i$-th row, $j$-th column visual patch of the $t$-th frame. This structured semantic grounding is not present in earlier fusion approaches.
>
> 3. **Avoiding redundancy**: As discussed in our response to Reviewer SwFQ's Q1, SyncFusion not only captures fine-grained AV synchronization but also avoids the redundancy issues seen in bidirectional or token-heavy fusion methods, further demonstrating its novelty and task-specific effectiveness.
>
> > **Q2**: If the temporal-spatial prior query is removed—keeping only JavisQuery and JavisCond—how much do synchrony and generation quality drop? Please provide an ablation isolating its contribution.
>
> Thansks for the comment. As shown in the table below, removing the temporal-spatial prior query has minimal impact on overall generation quality and consistency, but leads to a slight drop in audio-visual synchrony. This result aligns with our motivation: the temporal-spatial prior primarily serves to enhance the temporal and spatial alignment of generated audio and video content.
>
> | Strategy          |  Quality (V/A) ↓ |    Consistency (TV/TA/AV) ↑   | Synchrony ↑ |
> |--------------------|:---------:|:-----------------:|:---------:|
> | w/o ST-Prior Query | 319.3/**7.6** | **0.146**/0.178/**0.203** | 0.150     |
> | w/ ST-Prior Query  | **317.5**/**7.6** | 0.145/**0.180**/0.202 | **0.157**     |
>
>
> > **Q3**: Could the frozen DiT be replaced with a lighter autoregressive or VQ-GAN decoder without retraining the LLM adapters? Any preliminary experiments or rationale?
>
> **Ans**: Probably not. The latent space output by the LLM adapters must align with the input space of the decoder; switching from a frozen DiT to another autoregressive or VQ-GAN decoder would break this compatibility and cannot be applied directly. One possible workaround is to retrain the adapters, or alternatively, use an agent-based approach that first generates a text caption from the instruction and then feeds it to the decoder. However, this is not recommended. Although enabling quick adaptation and application, this paradigm hurts the unified nature of the entire framework.
>
> ---
>
> [1] Chen L, Wei X, Li J, et al. Sharegpt4video: Improving video understanding and generation with better captions[J]. Advances in Neural Information Processing Systems, 2024, 37: 19472-19495.
>
> [2] Yao L, Li L, Ren S, et al. Deco: Decoupling token compression from semantic abstraction in multimodal large language models[J]. arXiv preprint arXiv:2405.20985, 2024.

---

> > ### Comment · Reviewer_3pH1 · 2025-08-04
> >
> > I thank the authors for providing a detailed response. I will increase my score.
> >
> > Best wishes,

---

> > > ### Author Response · Authors · 2025-08-05
> > >
> > > Thank you for your kind response and score adjustment. We’re glad to hear that our rebuttal addressed your concerns. Thanks again for your insightful comments and constructive feedback, which further strengthen our work.

---

### Official Review · Reviewer_SwFQ · 2025-07-03

**Clarity:** 3
**Significance:** 3
**Originality:** 3
**Rating:** 4
**Confidence:** 3

**Summary:**

This article introduces JavisGPT, which is a unified MLLM specifically designed for audio-video coherent understanding and generation. It adopts an encoder-LLM-decoder architecture based on Qwen2.5-VL, BEAT, and pre-trained JAV-DiT generator, and bridges it through the SyncFusion module. Meanwhile, the paper constructs and uses JavisInst-Omni, which is an internal dataset consisting of 200K GPT-4o audio-video-text dialogues. In the understanding and generation benchmark tests of the paper, especially in the time synchronization task, JavisGPT's performance is significantly superior to that of previous MLLMs.

**Questions:**

- In addition to the concerns raised, the performance gains from SyncFusion appear relatively modest in table 4, and the reduced latency seems heavily tied to using fewer encoded AV tokens. This doesn’t convincingly show its superiority over Bidirectional Cross-Attention.
- Could the authors provide additional experiments or insights to clarify where SyncFusion outperforms Bidirectional Cross-Attention, both in accuracy and efficiency?
- Furthermore, I’m curious about how compressing audio-visual tokens impacts speed and other metrics. specifically:
Does cutting down on AV tokens significantly boost inference speed?
Are there hidden costs, like a drop in accuracy or other performance trade-offs?

**Ethical Concerns:**

["NO or VERY MINOR ethics concerns only"]

**Final Justification:**

I thank the authors for their comprehensive and timely rebuttal, which addressed my main concerns. The expansion of the human evaluation pool and the additional analysis on SyncFusion’s efficiency and accuracy provide valuable clarification. The updated results and further contextualization of the trade-offs strengthen the overall case for the paper. While there is still room for improvement in the scale of human evaluation and breadth of baseline comparisons, the technical merits and thoughtful design of the framework outweigh these limitations.

Given the clarified contributions and the added experimental evidence, I believe the paper makes a meaningful and relevant contribution to the field. I am maintaining my rating of borderline accept.

**Limitations:**

Yes

**Quality:**

3

**Strengths And Weaknesses:**

Strength
- This study introduces a new instruction dataset, which covers a wide range of real-world scenarios.
- SyncFusion effectively aligns fine‑grained audio and video features, enabling coherent sounding‑video generation from natural instructions.
- The three-stage training pipeline is well-conceived, offering a systematic way to incrementally build and refine the model's complex multimodal capabilities.

Weakness
- The human evaluation in this study involved only three volunteers, which may introduce bias due to the small sample size and limited diversity of assessors. Future work could benefit from a larger and more diverse group of evaluators to enhance the reliability and generalizability of the findings.
- The paper introduces heavy components like JAV-DiT and SyncFusion, which increase computational complexity and may hinder real-time or resource-constrained deployment. Though some comparisons with alternative methods are included, the scope is limited and does not cover simpler options like mlp. Thus, it remains unclear if the performance gains justify the higher computational costs and latency.

---

> ### Author Rebuttal · Authors · 2025-07-31
>
> > **W1**: The human evaluation in this study involved only three volunteers, which may introduce bias due to the small sample size and limited diversity of assessors. Future work could benefit from a larger and more diverse group of evaluators to enhance the reliability and generalizability of the findings.
>
> **Ans**: Thank you for the suggestion. During the rebuttal period, we added two additional volunteers with different backgrounds to the human evaluation. The updated averaged results are reported below, and the original conclusion holds, where JavisGPT consistently outperforms NExT-GPT and UnifiedIO-2 in both instruction-following and audio-visual understanding/generation capabilities. In future work, we plan to expand the evaluation along more dimensions and involve a more diverse group of assessors to further improve the robustness and generalizability of the human evaluation.
>
> | Model       | Instruction Following | Question Answering | Generation Quality | Context Reasoning | Proactive Thinking |
> |-------------|:---------------------:|:------------------:|:------------------:|:-----------------:|:------------------:|
> | UnifiedIO-2 |     74     |     71    |     58    |     63     |     52     |
> | NExT-GPT    |     52     |     74    |     44    |     69     |     59     |
> | JavisGPT    |     **94**     |     **90**    |     **88**    |     **80**     |     **81**     |
>
>
> > **W2**: The paper introduces heavy components like JAV-DiT and SyncFusion, which increase computational complexity and may hinder real-time or resource-constrained deployment. Though some comparisons with alternative methods are included, the scope is limited and does not cover simpler options like mlp. Thus, it remains unclear if the performance gains justify the higher computational costs and latency.
>
> **Ans**: Thanks for the comment. As discussed in Section 4, our core contribution lies in the overall framework design and optimization, where most components are modular and configurable to suit different deployment scenarios, including the sizes of the LLM and DiT backbones that dominate computational cost. The additional adaptors have limited impact on overall computation. For example, SyncFusion (CrossAttn+MLP) and standalone MLP layers introduce only 154.2M and 102.8M parameters respectively, accounting for just 1–2% of a 7B-scale LLM.
>
> In fact, the “Concat” baseline in Table 4 is functionally equivalent to an MLP-style adaptor, as any mapping from AV representations to the LLM input space requires some form of lightweight projection. The results show that SyncFusion achieves better performance with comparable or even lower computational overhead, justifying its inclusion in the system.
>
>
> |   Component   | AVQA Acc ↑  | MU-AVQA Acc ↑ | AVSD Acc ↑ | Param. Num. | Inference Latency ↓ |
> |:----------:|:--------:|:------------:|:--------:|:--------:|:-----------------:|
> | Concat (MLP) |   93.3   |     80.7     |   61.3   |   **102.8M** |     246ms       |
> | **SyncFusion** | **93.4** |   **81.4**   | **62.0** |  154.2M | **224ms**     |
>
> > **Q1**: In addition to the concerns raised, the performance gains from SyncFusion appear relatively modest in table 4, and the reduced latency seems heavily tied to using fewer encoded AV tokens. This doesn’t convincingly show its superiority over Bidirectional Cross-Attention.
>
> **Ans**: SyncFusion is designed to benefit both performance and efficiency:
>
> - Efficacy: The performance gains are meaningful. As shown in Table 4, replacing Interleaved concatenation with Bi-directional Cross-Attention actually leads to a drop in performance on AVQA and AVSD tasks. It suggests that simply adding more attention computation does not guarantee improvement, whereas SyncFusion brings gains by performing frame-level injection and alignment, which reflects its task-specific design and usefulness.
>
> - Efficiency: In the LLM era, latency is tightly coupled with token count. Given that the computational backbone (LLM and DiT) remains unchanged, reducing the number of AV tokens directly lowers the inference cost during the prefill stage during deployment.
>
> |   Component   | AVQA Acc ↑  | MU-AVQA Acc ↑ | AVSD Acc ↑ | Inference Latency ↓ |
> |:----------:|:--------:|:------------:|:--------:|:--------:|
> | Interleave |   93.3   |     80.6     |   61.6   |  555ms |
> | BiCrossAttn |   93.2   |     80.7     |   61.1   |  263ms |
> | **SyncFusion** | **93.4** |   **81.4**   | **62.0** | **224ms** |
>
> > **Q2**: Could the authors provide additional experiments or insights to clarify where SyncFusion outperforms Bidirectional Cross-Attention, both in accuracy and efficiency?
>
> **Ans**: The main limitation of Bidirectional Cross-Attention lies in **information redundancy**. Based on the results in Table 4, we can draw the following insights:
>
> - Accuracy: After mutual fusion, video tokens carry auditory information and audio tokens also carry visual information, making it harder for the model to distinguish which modality to rely on for specific questions. This increases the complexity of reasoning and leads to reduced answer accuracy.
>
> - Efficiency: Compared to SyncFusion, Bidirectional Cross-Attention retains a large number of enhanced audio tokens, which adds to the computational load and increases latency during inference.
>
> In summary, SyncFusion removes redundant audio tokens while preserving essential cross-modal alignment, thereby improving both accuracy and efficiency over Bidirectional Cross-Attention.
>
> > **Q3**: Furthermore, I’m curious about how compressing audio-visual tokens impacts speed and other metrics. specifically: Does cutting down on AV tokens significantly boost inference speed? Are there hidden costs, like a drop in accuracy or other performance trade-offs?
>
> **Ans**: Naively compressing audio-visual tokens can improve inference speed to some extent, but it often comes at a significant cost in performance. As shown in the table below, during rebuttal we conducted a simple experiment where AV tokens were compressed via 2× average pooling, and observed the following:
>
> - Inference speed showed only a minor improvement, where latency dropped from 246 ms to 206 ms. As discussed earlier, since AV token count only affects the prefill stage, the speedup is limited and not particularly significant.
>
> - Performance dropped noticeably, where AVSD accuracy particularly decreased by nearly 3%, indicating that naive compression leads to substantial information loss.
>
> These results are consistent with our original motivation: instead of compressing AV tokens directly, our SyncFusion strategy reduces token redundancy while preserving critical cross-modal information, achieving both speed and accuracy gains.
>
> |   Compression Strategy   | AVQA Acc ↑  | MU-AVQA Acc ↑ | AVSD Acc ↑ | AV Tokens | Inference Latency ↓ |
> |:----------:|:--------:|:------------:|:--------:|:--------:|:-----------------:|
> | None  |   93.3   |     80.7     |   61.3   |    3.5K    |     246ms       |
> | AvgPooling  |   91.9   |     78.4     |   58.6   |    **1.8K**    |    **206ms**       |
> | **SyncFusion** | **93.4** |   **81.4**   | **62.0** |  2.0K  |     224ms     |

---

> > ### Comment · Reviewer_SwFQ · 2025-08-06
> >
> > I would like to thank the authors for their comprehensive and timely responses to my comments and questions. The clarifications and additional experiments are appreciated and provide further insight into the strengths and trade-offs of the proposed JavisGPT framework.
> >
> > W1: I acknowledge and appreciate the authors' efforts to expand the pool of human evaluators during the rebuttal period. The updated results, with added diversity among assessors, address the primary concern about evaluation bias to a reasonable extent for the current submission. I look forward to seeing continued expansion and rigor in future evaluations, as the authors have indicated.
> >
> > W2: The clarification regarding the modularity of the framework and the quantification of parameter/latency overheads introduced by SyncFusion vs. MLP is helpful. Demonstrating that SyncFusion remains lightweight relative to the full model is reassuring. However, I still recommend that future work includes broader ablations with even more lightweight adaptors, particularly given deployment concerns for some users.
> >
> > Q1/Q2: The authors' explanation and new comparative results on accuracy and latency are informative. The analysis of token redundancy and modality confusion in Bidirectional Cross-Attention, along with the specific improvements in both accuracy and efficiency by SyncFusion, are convincing. The quantitative gains, though modest, are now better contextualized and the rationale for SyncFusion's design is clearer.
> >
> > Q3: The results regarding naive token compression are useful. The evidence that substantial performance is lost with only marginal latency gains supports the need for more sophisticated approaches like SyncFusion for effective token reduction.
> >
> > The authors have addressed my main concerns and provided thoughtful supplementary results. While some limitations remain—particularly regarding human evaluation scale and the need for broader comparison to simple baselines—the paper is technically sound and the added clarifications strengthen the overall case for acceptance.

---

> > > ### Author Response · Authors · 2025-08-07
> > >
> > > We greatly appreciate the reviewer's time and effort in reviewing our paper and rebuttal, and are glad to see that we have addressed all of the major concerns. The discussions and clarifications have strengthened our paper and will be integrated into the revised version.
> > >
> > > Best regards,
> > >
> > > Authors

---

### Official Review · Reviewer_MnnK · 2025-07-15

**Clarity:** 3
**Significance:** 4
**Originality:** 4
**Rating:** 6
**Confidence:** 4

**Summary:**

# Summary

This paper proposes a novel model for unified understanding and generation of synchronized audio-video. The model is built upon the Qwen2.5-VL-7B-Instruct backbone. For encoding, images are processed using Qwen2.5-VL’s visual encoder, while audio is encoded with BEATs. An additional SyncFusion module, based on cross-attention, is introduced to obtain synchronized audio-video features, with its output dimensionality aligned to that of the encoded video features. These fused features are then fed into the LLM backbone. For generation, multiple query tokens are provided to the LLM backbone, which then produces latent conditions for an external DiT module responsible for synchronized audio-video generation. The output conditions are aligned with the T5-XXL text encoder, and the DiT module remains frozen during training. The authors further propose a hierarchical query token approach, where the output conditions of these additional queries are aligned with the Spatial-Temporal Prior encoder of JavisDiT. The final loss function comprises four terms: (1) semantic alignment loss, (2) ST-prior alignment loss, (3) denoising objective (not used during pre-training), and (4) NTP objective. The model is trained in three stages: (1) a pre-training stage to align audio inputs and multimodal outputs, (2) an audio-video fine-tuning stage to enhance synchronized audio-video comprehension and generation, and (3) instruction-tuning to improve instruction-following and reasoning abilities. Additionally, the authors introduce the JavisInst-Omni dataset.

**Questions:**

1. **Lines 217–220**: This experimental finding is quite interesting. I am very curious about it. Generally, for video generation models, a two-stage approach involving prompt engineering (PE) and DiT is used: the PE stage leverages a language model or MLLM to generate a caption for the target video, and then DiT uses a text encoder to obtain the conditions for video generation. Although some recent methods have attempted to bypass the PE stage by directly generating such conditions with an LLM, none have shown superior video generation performance compared to the traditional PE+DiT approach. Therefore, the finding in this paper that JavisGPT outperforms JavisDiT is particularly intriguing. Can you provide more insights into this? What are the key factors that allow the latter solution to surpass the PE+DiT approach? How do you ensure that, even with only a vague textual condition (and not a clear target video caption), the performance of direct condition generation can exceed that of PE+DiT? Or what are the reasons for which the previous non-PE approaches fail?

2. **Lines 109–114**: The authors freeze JavisDiT for the sake of training efficiency and simplicity. What would happen if JavisDiT were also trained (i.e., unfrozen)? Could this potentially further improve performance? What are your thoughts on the possible training stability issues when unfreezing JavisDiT, especially in terms of avoiding significant degradation in DiT performance? (I know that JavisDiT is frozen in this work, but I am very curious about this.)

3. When will this model be open-sourced?

This paper is impressive. I am willing to increase my score if the weaknesses and my questions are addressed.

**Ethical Concerns:**

["NO or VERY MINOR ethics concerns only"]

**Final Justification:**

The authors have addressed most of my concerns and questions.

**Limitations:**

Yes.

**Paper Formatting Concerns:**

None.

**Quality:**

4

**Strengths And Weaknesses:**

# Strengths

1. This work represents a significant milestone in the field of sounding video understanding and generation.
2. The technical contributions are substantial and introduce meaningful innovations.
3. The experiments are well-designed, and the findings are both solid and interesting.
4. JavisGPT demonstrates impressive performance.
5. The ablation studies are thorough and provide valuable insights.

# Weaknesses

1. Why did the authors choose BEATs (2022) as the audio encoder, instead of adopting more recent advancements such as WavTokenizer (2024)[1]?
2. It would be beneficial to provide more details about the evaluation metrics for audio-video generation, perhaps in the appendix.
3. What are the benefits of cross-attention mechanism (SyncFusion) compared with interleaved concatenation ([Vclip0, Aclip0, Vclip1, Aclip1, ...]) for audio-video synchrony?

[1] wavtokenizer: an efficient acoustic discrete codec tokenizer for audio language modeling

---

> ### Author Rebuttal · Authors · 2025-07-31
>
> We sincerely thank the reviewer for the recognition and valuable insights, which have also been inspiring to us. We hope the following responses will help address the remaining concerns.
>
> > **W1**: Why did the authors choose BEATs (2022) as the audio encoder, instead of adopting more recent advancements such as WavTokenizer (2024)?
>
> **Ans**: There are several main reasons:
> 1. We follow prior works (e.g., Video-Salmonn[1], LongVALE-LLM[2]) in using BEATs, due to its strong generalization ability in audio representation.
> 2. BEATs is optimized for sound effects, which aligns well with the focus of our work. Differently, WavTokenizer is primarily designed for speech processing (although it also supports music and audio), which is not well aligned with the focus of our study.
>
> In future, we are happy to explore WavTokenizer for speech understanding or generation in particular.
>
> > **W2**: It would be beneficial to provide more details about the evaluation metrics for audio-video generation, perhaps in the appendix.
>
> **Ans**: Thank you for the suggestion. We have provided a brief explanation of the metrics used to evaluate audio-video quality, consistency, and synchrony in the supplementary material (appendix.pdf, Section B.3), including their definitions and reference computation methods. We will include more detailed descriptions of the calculation procedures in the revised paper to improve readability.
>
> > **W3**: What are the benefits of cross-attention mechanism (SyncFusion) compared with interleaved concatenation ([Vclip0, Aclip0, Vclip1, Aclip1, ...]) for audio-video synchrony?
>
> **Ans**: SyncFusion offers both performance and efficiency benefits, as supported by quantitative results below (partially copied from Table 4 in the paper):
> - In terms of performance, SyncFusion injects audio information directly into visual frames, which reduces the burden on the model to capture synchrony across separate audio and video tokens, thereby improving comprehension and accuracy.
> - In terms of efficiency, SyncFusion reduces the number of tokens to process and avoids interleaving audio and video tokens, significantly reducing memory overhead and lowering inference latency.
>
> |   Mechanism   | AVQA Acc ↑  | MU-AVQA Acc ↑ | AVSD Acc ↑ | Inference Latency ↓ |
> |:----------:|:--------:|:------------:|:--------:|:-----------------:|
> | Interleave |   93.3   |     80.6     |   61.6   |        555ms       |
> | **SyncFusion** | **93.4** |   **81.4**   | **62.0** |    **224ms**     |
>
> > **Q1**: Lines 217–220: The finding that JavisGPT outperforms JavisDiT is particularly intriguing. Can you provide more insights into this? What are the key factors that allow the latter solution to surpass the PE+DiT approach? How do you ensure that, even with only a vague textual condition (and not a clear target video caption), the performance of direct condition generation can exceed that of PE+DiT? Or what are the reasons for which the previous non-PE approaches fail?
>
> **Ans**:
> First, we would like to clarify that in the experimental setting of Lines 217–220 (Table 3), both JavisDiT and JavisGPT receive the same detailed textual input, without involving additional prompt engineering process. Compared to the standard text encoder (T5) used in DiT, JavisGPT leverages an generative LLM (Qwen) to integrate information, which offers stronger semantic understanding and reasoning capabilities. For example, given the instruction "a cat jumps onto a table and knocks over a cup", a conventional encoder tends to process tokens like “cat”, “table”, and “cup” as discrete elements. In comparison, the LLM captures the causal and temporal relationship between actions, inferring that jumping onto the table leads to knocking over the cup. This supports the generation of more coherent and logically structured videos. This design philosophy aligns with recent works like Hunyuan-Video[3], where LLMs are also used as semantic extractors to guide generation.
>
> Regarding vague textual conditions, traditional PE+DiT pipelines rely on an external LLM to expand user input into a detailed caption. However, the style and distribution of such generateds caption may differ from the text that DiT has seen during training, including phrasing, sentence structure, and detail granularity. This creates a style gap that hinders generation quality. Differently, JavisGPT jointly trains the LLM and DiT modules, effectively reducing this mismatch. During inference, JavisGPT proactively completes the vague instruction into a richer caption, and upon user confirmation, passes the full semantic context to the DiT decoder via JavisQuery for sounding video generation. This mechanism is illustrated in Appendix Figure A3 (Rethinking Generation), which shows how our framework handles vague instructions while maintaining semantic fidelity.
>
> > **Q2**: Lines 109–114: The authors freeze JavisDiT for the sake of training efficiency and simplicity. What would happen if JavisDiT were also trained (i.e., unfrozen)? Could this potentially further improve performance? What are your thoughts on the possible training stability issues when unfreezing JavisDiT, especially in terms of avoiding significant degradation in DiT performance? (I know that JavisDiT is frozen in this work, but I am very curious about this.)
>
> **Ans**: Great question. We would draw your attention to the observation in MetaQuery[4], where unfreezing DiT and training it jointly with other components could theoretically lead to better performance. However, there are several practical challenges that must be addressed:
>
> 1.  **Training stability** is the primary concern. In fact, even with the DiT frozen, if we skip the MM-PT (Multimodal Pretraining) stage and directly connect the modules using diffusion loss, the system tends to completely break, rather than merely suffering from “significant degradation” in DiT performance. Therefore, if we were to train JavisDiT jointly, we would need to adopt a multi-stage training strategy or apply advanced stabilization techniques such as learning rate scheduling and gradient clipping.
>
> 2. Another key constraint is **memory usage**. JavisDiT has over 3 billion parameters, and unfreezing it would add approximately 48GB (16 × 3GB) of GPU memory consumption. This would require higher-level ZeRO parallelism, parameter offloading, and other advanced memory optimization strategies, which would significantly increase the engineering complexity of training.
>
> That said, we believe that with extensive optimization, jointly training JavisDiT could indeed lead to further performance gains, and we plan to explore this direction in future work.
>
> > **Q3**: When will this model be open-sourced?
>
> **Ans**: We are currently organizing all the code, data, and model checkpoints, and plan to open-source the project as soon as possible.
>
> ---
>
> [1] Sun G, Yu W, Tang C, et al. video-SALMONN: speech-enhanced audio-visual large language models[C]//Proceedings of the 41st International Conference on Machine Learning. 2024: 47198-47217.
>
> [2] Geng T, Zhang J, Wang Q, et al. Longvale: Vision-audio-language-event benchmark towards time-aware omni-modal perception of long videos[C]//Proceedings of the Computer Vision and Pattern Recognition Conference. 2025: 18959-18969.
>
> [3] Kong W, Tian Q, Zhang Z, et al. Hunyuanvideo: A systematic framework for large video generative models[J]. arXiv preprint arXiv:2412.03603, 2024.
>
> [4] Pan X, Shukla S N, Singh A, et al. Transfer between modalities with metaqueries[J]. arXiv preprint arXiv:2504.06256, 2025.

---

> ### Comment · Reviewer_MnnK · 2025-08-05
> **Responses to the Rebuttal**
>
> W1:
> I believe that employing a more advanced audio encoder, such as Whisper, could further enhance performance. Although Whisper is primarily trained on speech data, it has demonstrated strong performance on both general sounds and music (see Qwen-Omni for reference).
>
> W2:
> Providing a more detailed description of the evaluation metrics would improve the clarity of this paper. For example, including the evaluation procedure for JavisScore within the paper would save readers the time required to search the reference paper.
>
> W3:
> In my opinion, the main advantage of SyncFusion lies in its efficiency rather than its performance. The performance improvement is relatively limited.
> Moreover, JavisScore is designed for evaluating Audio-Video Synchrony. What's the comparison between interleaved concatenation and SyncFusion using this metric?
>
> Q1:
> What if the model is trained with generated captions (PE captions)? Can I see that PE feature→DiT is preferable to PE caption→Qwen Encoder→DiT due to the reduced gap between training and inference, even if both approaches are trained with rigorous alignment (for example, training the Qwen Encoder+DiT on PE captions rather than other captions)? Note that PE features refer to Qwen-encoded user input. Moreover, is it necessary for the Qwen Encoder to be trained with an additional loss function, PE loss, which refers to the NTP loss for generating the target dense caption?

---

> > ### Author Response · Authors · 2025-08-05
> >
> > Thank you very much for your thoughtful and detailed feedback.
> >
> > - W1: We appreciate the suggestion of trying more recent audio encoders in the framework. We agree it is a promising direction and will consider it in future extensions, such as trying separate sound, speech, and music encoders or an integrated encoder to further enhance the audio representation.
> >
> > - W2: Thank you for pointing this out. All the evaluation metrics for sounding video generation follow JavisDiT[1], including the JavisScore metric (basically measuring the segment-level consistency between the whole audio-video pairs). We will add a more detailed explanation of the evaluation metrics in the revised version to improve clarity and self-containment.
> >
> > - W3: In fact, JavisScore is designed to evaluate the audio-visual synchrony in generated sounding videos, while interleaved concatenation and SyncFusion modules are designed to caption the synchrony in input sounding videos for better understanding. Therefore, JavisScore cannot be adopted to evaluate interleaved concatenation and SyncFusion modules, and understanding metrics are more suitable for evaluation, such as the QA accuracy used in our experiments.
> >
> > - Q1: Yes, we argue that using PE features (i.e., Qwen-encoded user inputs) aligns training and inference more closely, reducing modality and distribution gaps. And adding a PE/NTP loss on the target dense caption for training is indeed necessary, since the inference stage will force the LLM to generate target captions and then pass the context information into the DiT decoder via JavisQuery. We will continue to explore this in future work, such as developing a more advanced thinking model version to adapt to this situation (generating dense captions can be viewed as a thinking process) and further enhance both understanding and generation of sounding videos.
> >
> > Thanks again for your constructive comments, which are very helpful in strengthening our paper.
> >
> > ---
> >
> > [1] Liu K, Li W, Chen L, et al. Javisdit: Joint audio-video diffusion transformer with hierarchical spatio-temporal prior synchronization[J]. arXiv preprint arXiv:2503.23377, 2025.

---

> > > ### Comment · Reviewer_MnnK · 2025-08-05
> > > **Responses to the Author**
> > >
> > > Thank you for the rebuttal. I am willing to increase my score.

---

> > > > ### Author Response · Authors · 2025-08-05
> > > >
> > > > Thank you for your positive feedback and for engaging in the constructive discussion period, which helped us improve the clarity and quality of our paper. All the discussions will be integrated into the revised version.
> > > >
> > > > Best regards,
> > > >
> > > > Authors

---

### Decision · Program_Chairs · 2025-09-17

**Decision:**

Accept (spotlight)

**Comment:**

The paper proposes a new MLLM for synchronized AV understanding and generation based on strong spatio-temporal alignment. It proposes a  three-stage training pipeline for synchronized generation as well as a new large-scale dataset for complex audio-video-text interactions based on QA instances from existing models such as VideoLlama2. The model shows good results across various benchmarks.

The paper was considered by four reviewers with the following ratings: SA - BA - BA - BA

The reviewers mainly criticised the limited conceptual novelty, but highlighted the extensive evaluation, the solid model design, as well as the option of having access to this as a public model and dataset.  Most weaknesses raised by reviewers have been addressed in the rebuttal.

The AC follows the consensus of the reviewer voting and recommends accepting the paper.
The AC would encourage the authors to integrate the findings of the rebuttal in the CR version of the paper.